# Artificial Neural Network analysis on the effect of mixed convection in triangular-shaped geometry using water-based Al$_2$O$_3$ nanofluid

**M. N. Hudha** [1], **Md. Jahid Hasan** [2], **T. Bairagi** [1], **A. K. Azad** [3], **M. M. Rahman** [1] *

**1** Department of Mathematics, Bangladesh University of Engineering and Technology, Dhaka, Bangladesh, **2** Department of Mechanical and Production Engineering, Islamic University of Technology, Gazipur, Bangladesh, **3** Department of Natural Sciences, Islamic University of Technology, Gazipur, Bangladesh

* m71ramath@gmail.com

**Data Availability Statement:** All relevant data are within the manuscript.

**Funding:** The author(s) received no specific funding for this work.

## Abstract

The objective of the study is to investigate the fluid flow and heat transfer characteristics applying Artificial Neural Networks (ANN) analysis in triangular-shaped cavities for the analysis of magnetohydrodynamics (MHD) mixed convection with varying fluid velocity of water/Al$_2$O$_3$ nanofluid. No study has yet been conducted on this geometric configuration incorporating ANN analysis. Therefore, this study analyzes and predicts the complex interactions among fluid flow, heat transfer, and various influencing factors using ANN analysis. The process of finite element analysis was conducted, and the obtained results have been verified by previous literature. The Levenberg-Marquardt backpropagation technique was selected for ANN. Various values of the Richardson number ($0.01 \leq$ Ri $\leq 5$), Hartmann number ($0 \leq$ Ha $\leq 100$), Reynolds number ($50 \leq$ Re $\leq 200$), and solid volume fraction of the nanofluid ($\phi$ = 1%, 3% and 4%) have been selected. The ANN model incorporates the Gauss-Newton method and the method of damped least squares, making it suitable for tackling complex problems with a high degree of non-linearity and uncertainty. The findings have been shown through the use of streamlines, isotherm plots, Nusselt numbers, and the estimated Nusselt number obtained by ANN. Increasing the solid volume fraction improves the rate of heat transmission for all situations with varying values of Ri, Re, and Ha. The Nusselt number is greater with larger values of the Ri and Re parameters, but it lessens for higher value of Ha. Furthermore, ANN demonstrates exceptional precision, as evidenced by the Mean Squared Error and R values of 1.05200e-6 and 0.999988, respectively.

## 1. Introduction

Mixed convective fluid flow and heat transfer in different enclosures considering various conditions has been a notable area of research in the past decade. Various models and methodologies are involved in order to incorporate multiple shapes, fluids, and boundary conditions. Raza et al. [1] investigated the fluid dynamics of ternary solid particles, nano-layers, and the impact of magnetic fields in the presence of porous disks. Their research centered on

**Competing interests:** On behalf of all authors, I declared that we have no known competing financial interests or personal relationships that could have appeared to influence the work reported in this paper.

**Abbreviations:** $C_p$, Specific heat at constant pressure ($Jkg^{-1}K^{-1}$); Gr, Grashoff number; H, Convective heat transfer coefficient ($Wm^{-2}K^{-1}$); Ha, Hartmann number; k, Thermal conductivity of the material ($Wm^{-1}K^{-1}$); L, Length of the enclosure (m); p, Dimensional pressure ($Nm^{-2}$); P, Dimensionless pressure; Pr, Prandtl number; Re, Reynolds number; Ri, Richardson number; T, Dimensional temperature (K); u, v, Dimensional velocity components ($ms^{-1}$); U, V, Dimensionless velocity components; x, y, Dimensional cartesian coordinates; X, Y, Dimensionless cartesian coordinates; $B_0$, Magnetic field induction (T); $\psi$, Stream function;
**Greek Symbols** $\alpha$, Thermal diffusivity ($m^2s^{-1}$); $\beta$, Co-efficient of thermal expansion ($K^{-1}$); $\theta$, Dimensionless temperature; $\mu$, Dynamic viscosity of the fluid ($kgm^{-1}s^{-1}$); $\rho$, Density of the material ($kgm^{-3}$); $\sigma$, Electrical conductivity ($\Omega^{-1}m^{-1}$); $\upsilon$, Kinematic viscosity of the fluid ($m^2s^{-1}$); $\phi$, Solid volume fraction;
**Subscripts** avg, average; bf, base fluid; c, cold; h, hot; nf, nanofluid; sp, solid particle.

improving heat transfer by considering the kinetics of entropy formation and factors such as joule heating effects and chemical interaction with nanoparticles. Irshad et al. [2] investigated the thermal characteristics and entropy analysis of double-diffusive phase transition materials within a porous H-shaped wavy cavity. Utilizing the Forchheimer-Brinkman enlarged Darcy medium model, they investigated the impact of several key points on convective heat and mass transmission.

Magnetohydrodynamics (MHD) or magnetic field is often considered in the thermal problems for specific applications, as it has a significant impact on hydrothermal performance [3–7]. Selimefendigil and Öztop [8] investigated the phase transition and thermal processes occurring in a wavy enclosure. The enclosure contained a spherical-shaped encapsulated phase change material (PCM) and was subjected to magneto-convection of nanofluid. Active rotations at Re = 100 resulted in a 12.5% decrease in phase transition time, but at Re = 0, there was a 16.6% increase. Mahabaleshwar et al. [9] studied the flow of power-law nanofluids across a decreasing sheet under MHD. They derived analytical solutions that demonstrated the influence of the Chandrasekhar number, nanofluid, and inclination angle parameters on the flow dynamics. The study discovered that the magnetic field substantially impacted skin friction and mass transpiration. Khalili et al. [10] investigated the augmentation of electrical power in solar panels by introducing $Fe_3O_4$ nanoparticles into water. The combination of magnetic field-assisted cooling and a thermoelectric generator resulted in a 5.07% improvement in performance as the magnetic field intensity (Ha) increased and an 11.70% improvement with higher inlet velocity.

Nanofluid has been extensively used in various studies to enhance thermal performance [11–13]. Porgar et al. [14] conducted a study on the use of carbon nanotubes in transformer oil nanofluids. The results revealed that the effective thermal conductivity of the nanofluids augmented as the temperature and concentration of carbon nanotubes increased. The maximum enhancement was observed at a temperature of 45°C and a volume fraction of 0.56. Rajesh and Öztop [15] investigated the effects of a ternary hybrid nanofluid on heat transfer in a square-enclosed environment using numerical simulations. It provided extensive assessments of average Nusselt numbers and profiles of streamlines and isotherms while considering varied nanoparticle volume percentages and Rayleigh numbers. Belabid and Öztop [16] analyzed the thermal process in a cylindrical porous annulus with wavy walls, where phototactic microorganisms and hybrid nanofluid were present. The results indicated that the Lewis number significantly impacts thermo-convection, nanoparticles improve heat transmission, and increasing wall waviness characteristics lead to a drop in the Nu. Sheikholeslami and Jafaryar [17] simulated turbulent flow in a solar system absorber tube using a swirl flow device, including CNT-water nanomaterial. Gangadhar et al. [18] examined the properties of Casson-Maxwell nanofluid between stationary discs, taking into account thermal radiation and chemical interactions. The results demonstrated that the temperature improved as the Brownian motion and radiation parameters rose, and the rate of reaction impacted the system.

Shao et al. [19] utilized Response Surface Methodology (RSM), an advanced statistical technique, to optimize heat transfer in a micropolar nanofluid flow within a squeezing channel. This optimization process took into account many restrictions and parameters, such as the magnetic field and thermal radiation. The study found that a higher volume fraction increased fluid viscosity, which in turn caused a decrease in velocity. Jafaryar et al. [20] suggested an innovative configuration of a twisted tape integrated with helical fins heat sink to improve the convective flow within the absorber pipe of a solar water heater. The results indicated that including helical fins led to a 70.52% increase in Nu. Additionally, when the Reynolds number grew from 4500 to 9000, Nu increased by 76.31%, while the Darcy factor reduced by 22.72%. Selimefendigil and Öztop [21] studied how a magnetic field and revolving cylinders affect

thermal processes in a T-shaped channel. A zone filled with PCM was studied to determine how many things affected it. The lack of a magnetic field reduced transition time by 42.5% at Reynolds number 2000. Khalili and Sheikholeslami [22] proposed an innovative approach to cool photovoltaic (PV) panels by integrating them with a thermoelectric generator and utilizing a heat transfer tube combined with a restricted jet impingement heat sink. The cooling fluid consisted of water combined with hybrid nano-powders (ND-$Co_3O_4$) in different proportions. Under certain conditions, the modified system exhibited electrical and thermal efficiencies of around 84% and 15.44%, respectively, surpassing scenarios with lower sun irradiation. In another study [23], they suggested an improved cooling system for Photovoltaic (PV) cells by employing a numerical method that integrates a thermoelectric layer and a hybrid nanofluid in cooling ducts of different forms and Y-shaped fins. The triangular duct with fins and jets exhibited superior performance compared to a circular pipe, achieving a performance improvement of 9.97%.

In recent studies, Artificial Neural Networks have been incorporated in order to optimize the problems related to enclosure and cavity flow, considering heat transfer problems under mixed convection [24–28]. Çolak [29] examined the impact of viscous dissipation on heat transport over a stretchy surface using Cu-PVA Jeffrey nanofluid under the influence of magnetohydrodynamics. The study utilized ANN models that demonstrated a high level of predicted accuracy. The R values for skin friction and Nu were determined to be 0.99020 and 0.99394, respectively, indicating the model's high degree of reliability. Li et al. [30] presented a novel design for a microchannel heat sink that resembles a tree structure. This design was developed utilizing ANN and RSM. The predictions made by this design were highly accurate, with modified $R^2$ values reaching up to 0.997. Jery et al. [31] performed a simulation of a geothermal heat exchanger, where the goal was to reduce entropy generation by optimizing the diameter and nanoparticle concentration. The study found that enhancing the heat resistance of the inner wall in larger diameters reduced the impact of entropy, whereas the inclusion of nanoparticles resulted in an approximately 10% rise in the average Nusselt number. Cho et al. [32] used numerical methods to assess 3D mixed convection with different aspect ratios (AR). Higher AR increased heat transfer rates. The study showed that an ANN could predict heat transfer characteristics as accurately as DNS data. Filali et al. [33] demonstrated the viability of employing ANN for forecasting thermal dynamics in mixed convection, which was trained using numerical simulations of a square cavity with internal blocks that were driven by a lid. This training allowed the ANN to make accurate predictions for various combinations of Reynolds, Grashoff, and Richardson numbers and block distances. The results emphasized the impact of the spacing between blocks on the Nu, and specific correlations were developed for potential uses in engineering design. Alqaed et al. [34] utilized the SIMPLE algorithm to examine the natural convection of water-$Al_2O_3$ nanofluids in a cavity with triangular blades. The analysis uncovered a direct relationship between the Ra and both the Nu and the rate of entropy production. Nevertheless, the influence of the Ha on these variables was contingent upon the values of Ra.

To this point, numerous studies have been conducted to investigate fluid flow and thermal performance in enclosed flows. However, there is a significant research gap that needs to be filled. This study aims to fill the research gap by using Artificial Neural Networks analysis in triangular-shaped cavities for magnetohydrodynamics (MHD) mixed convection with water/$Al_2O_3$ nanofluid. The introduction of ANN analysis attempts to offer a new method for comprehending the intricate relationships among fluid flow, heat transmission, and the factors that affect these cavities. This research addresses a gap in the existing literature and adds to the overall body of research by providing precise predictions and insights for improving triangular-shaped cavities under mixed convection with magnetohydrodynamics. This is especially

pertinent for engineering applications and corresponds to the increasing interest in advanced computational methods for intricate fluid flow and heat transfer issues. Therefore, this study assesses the fluid flow, heat transfer, and conduct an Artificial Neural Network analysis for various Ri, Ha, and Re within the specified intervals of $0.01 \leq Ri \leq 5$, $50 \leq Re \leq 200$, and $0 \leq Ha \leq 100$, respectively. Additionally, the study takes into account the three-variable nanofluid concentrations (1%, 3% and 4%) for the water-based $Al_2O_3$ nanofluid. The heat transfer and thermal science community will greatly benefit from the outcomes of this research.

The following sections of the study will detail the approach used, which includes finite element analysis and the Levenberg-Marquardt backpropagation technique for artificial neural network analysis. The study will also discuss the chosen parameters that were taken into account, including the Ri, Ha, Re, and volume fraction. It will also offer the findings and conclusions gained in order to fill the research gap and enhance understanding in this field.

## 2. Fluid domain and mathematical formulation

### 2.1. Problem specifications

The physical model considers a uniformly charged $Al_2O_3$-$H_2O$ nanofluid in a triangular–shaped cavity with a two-dimensional, laminar, and incompressible mixed convection flow. In this study, alumina ($Al_2O_3$) is used as a nanoparticle, and water ($H_2O$) is used as the base fluid. The enclosure's height, H and base wall length, L = 2.40H, are represented by these letters. Additionally, for $0 < x < 2.40$, the zeroth order Bessel's function ($J_0(x)$) forms the shape of the curve. A mixed convection flow has been introduced by the moving wall. The curve-inclined wall warms up at low heat, $T = T_c$, while the bottom wall warms up at $T = \sin\left(\frac{\pi x}{L}\right)$. The insulation is retained on the vertical wall. The vertical wall symbolizes the y-coordinate, and the bottom wall symbolizes the x-coordinate. The direction of action of gravitational acceleration is negative. It is believed that there is no dynamical or thermal slip between the base fluid and the nanoparticles since they are uniformly distributed throughout it. For all solid barriers, no-slip walls are taken into account. The average Nu at the heated wall is investigated using water-based alumina ($Al_2O_3$) nanofluid. The fluid domain is illustrated in Fig 1, and Table 1 displays the thermophysical characteristics, including base fluids.

### 2.2. Mathematical and governing equations

**2.2.1 Dimensional governing equations.** The dimensional variant of the continuity, momentum under the Boussineq approximation, and energy equations for the 2-D mixed convective Newtonian fluid, laminar flow, and steady-state flows are as follows:

$$\frac{\partial u}{\partial x} + \frac{\partial v}{\partial y} = 0 \tag{1}$$

$$u\frac{\partial u}{\partial x} + v\frac{\partial u}{\partial y} = -\frac{1}{\rho_{nf}}\frac{\partial p}{\partial x} + \upsilon_{nf}\left(\frac{\partial^2 u}{\partial x^2} + \frac{\partial^2 u}{\partial y^2}\right) \tag{2}$$

$$u\frac{\partial v}{\partial x} + v\frac{\partial v}{\partial y} = -\frac{1}{\rho_{nf}}\frac{\partial p}{\partial y} + \upsilon_{nf}\left(\frac{\partial^2 v}{\partial x^2} + \frac{\partial^2 v}{\partial y^2}\right) + \frac{(\rho\beta)_{nf}}{\rho_{nf}}g(T - T_c) - \frac{\sigma_{nf}B_0^2}{\rho_{nf}}v \tag{3}$$

$$u\frac{\partial T}{\partial x} + v\frac{\partial T}{\partial y} = \alpha_{nf}\left(\frac{\partial^2 T}{\partial x^2} + \frac{\partial^2 T}{\partial y^2}\right) \tag{4}$$

Where due to significant temperature variations, density fluctuations related to temperature in

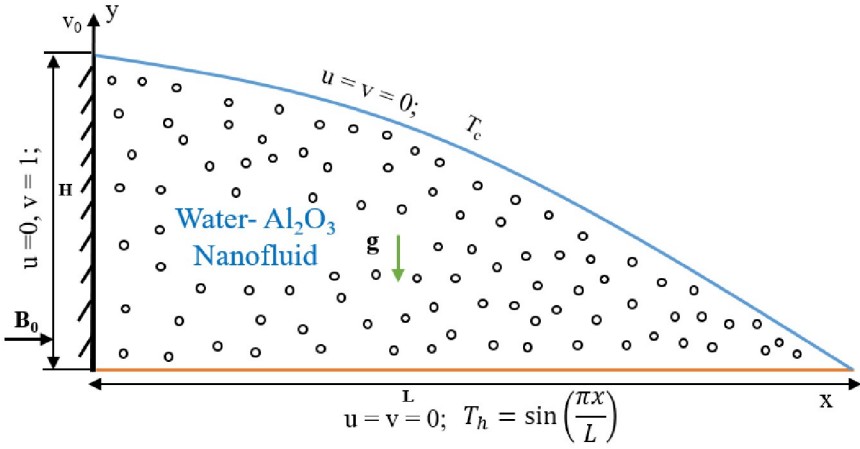

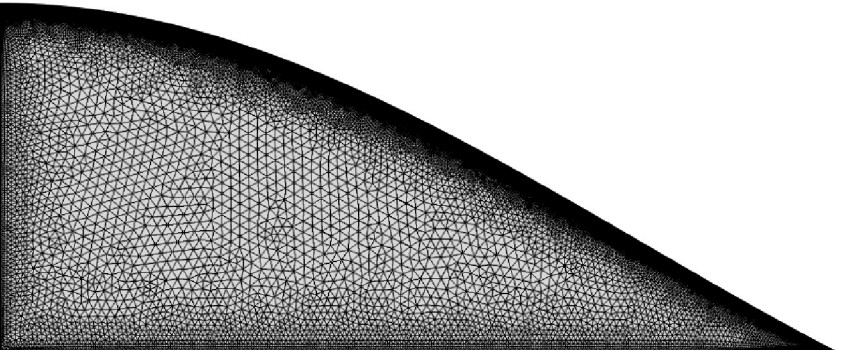

**Fig 1. Schematic graph and mesh view of proposed problem.**

the buoyancy term are approximated using a Boussinesq model. This means that density changes are assumed to follow a relationship with temperature $-\rho_0\beta(T - T_c)g$.

The boundary conditions for the given problem are as follows.

$$\left.\begin{array}{l}\text{At the vertical boundary}: u = 0, \ v = v_0, \ \dfrac{\partial T}{\partial n} = 0 \\[2mm] \text{At the horizontal boundary}: u = 0, \ v = 0, \ \mathrm{T} = T_h = \sin\left(\dfrac{\pi x}{L}\right) \\[2mm] \text{At the curve boundary}: u = 0, \ v = 0, \ T = T_c \end{array}\right\} \quad (5)$$

**Table 1. Thermophysical properties of base fluid and nanoparticle [Roslan et al. [35]].**

| Properties | Base fluid (Water) | Solid Nanoparticle ($AL_2O_3$) | Unit |
|---|---|---|---|
| $\rho$ | 997.1 | 3970 | $kgm^{-3}$ |
| $C_p$ | 4179 | 765 | $Jkg^{-1}K^{-1}$ |
| $k$ | 0.613 | 40 | $Wm^{-1}K^{-1}$ |
| $\beta$ | $21\times10^{-5}$ | $85\times10^{-7}$ | $K^{-1}$ |
| $\mu$ | $1.002\times10^{-6}$ | - | $kgm^{-1}s^{-1}$ |
| $\sigma$ | 0.05 | $1\times10^{-10}$ | $m^2s^{-1}$ |

**2.2.2 Dimensionless governing equations.** To formulate the Eqs (1)–(4) in a dimensionless format about their initial and boundary conditions Eq (5). The dimensionless factors are:

$$\left. \begin{aligned}
&X = \frac{x}{H}, \ Y = \frac{y}{H}, \ U = \frac{u}{v_0}, \ V = \frac{v}{v_0}, \ P = \frac{p}{\rho_{bf} v_0^2}, \ \theta = \frac{T - T_c}{T_h - T_c}, \\
&\Pr = \frac{v_{bf}}{\alpha_{bf}}, \ Gr = \frac{g \beta_{bf}(T_h - T_c)H^3}{v_{bf}^2}, \ \mathrm{Re} = \frac{v_0 H}{v_{bf}}, \ Ri = \frac{Gr}{\mathrm{Re}^2}, \ \text{and} \ Ha = B_0 H \sqrt{\frac{\sigma_{bf}}{\rho_{bf} v_{bf}}}
\end{aligned} \right\}$$

(6)

$$\left. \begin{aligned}
&\therefore \frac{\partial u}{\partial x} = \frac{v_0}{H}\frac{\partial U}{\partial X}, \frac{\partial u}{\partial y} = \frac{v_0}{H}\frac{\partial U}{\partial Y}, \frac{\partial v}{\partial x} = \frac{v_0}{H}\frac{\partial V}{\partial X}, \frac{\partial v}{\partial y} = \frac{v_0}{H}\frac{\partial V}{\partial Y}, \\
&\frac{\partial p}{\partial x} = \frac{\rho_{bf} v_0^2}{H}\frac{\partial P}{\partial X}, \frac{\partial p}{\partial y} = \frac{\rho_{bf} v_0^2}{H}\frac{\partial P}{\partial Y}, \frac{\partial T}{\partial x} = \frac{\Delta T}{H}\frac{\partial \theta}{\partial X}, \frac{\partial T}{\partial y} = \frac{\Delta T}{H}\frac{\partial \theta}{\partial Y}
\end{aligned} \right\}$$

(7)

$$\left. \begin{aligned}
&\therefore \frac{\partial^2 u}{\partial x^2} = \frac{v_0}{H^2}\frac{\partial^2 U}{\partial X^2}, \frac{\partial^2 u}{\partial y^2} = \frac{v_0}{H^2}\frac{\partial^2 U}{\partial Y^2}, \frac{\partial^2 v}{\partial x^2} = \frac{v_0}{H^2}\frac{\partial^2 V}{\partial X^2}, \\
&\frac{\partial^2 v}{\partial y^2} = \frac{v_0}{H^2}\frac{\partial^2 V}{\partial Y^2}, \frac{\partial^2 T}{\partial x^2} = \frac{\Delta T}{H^2}\left(\frac{\partial^2 \theta}{\partial X^2}\right), \frac{\partial^2 T}{\partial y^2} = \frac{\Delta T}{H^2}\left(\frac{\partial^2 \theta}{\partial Y^2}\right)
\end{aligned} \right\}$$

(8)

Applying the Eqs (6)–(8), the continuity, momentum, and energy equations- the dimensionless equations are as follows:

$$\frac{\partial U}{\partial X} + \frac{\partial V}{\partial Y} = 0$$

(9)

$$U\frac{\partial U}{\partial X} + V\frac{\partial U}{\partial Y} = -\frac{\rho_{bf}}{\rho_{nf}}\frac{\partial P}{\partial X} + \frac{v_{nf}}{v_{bf}}\frac{1}{\mathrm{Re}}\left(\frac{\partial^2 U}{\partial X^2} + \frac{\partial^2 U}{\partial Y^2}\right)$$

(10)

$$U\frac{\partial V}{\partial X} + V\frac{\partial V}{\partial Y} = -\frac{\rho_{bf}}{\rho_{nf}}\frac{\partial P}{\partial Y} + \frac{v_{nf}}{v_{bf}}\frac{1}{\mathrm{Re}}\left(\frac{\partial^2 V}{\partial X^2} + \frac{\partial^2 V}{\partial Y^2}\right) + \frac{(\rho\beta)_{nf}}{\rho_{nf}\beta_{bf}}Ri\theta - \frac{\rho_{bf}}{\rho_{nf}}\frac{\sigma_{nf}}{\sigma_{bf}}\frac{Ha^2}{\mathrm{Re}}V$$

(11)

$$U\frac{\partial \theta}{\partial X} + V\frac{\partial \theta}{\partial Y} = \frac{\alpha_{nf}}{\alpha_{bf}}\frac{1}{\mathrm{RePr}}\left(\frac{\partial^2 \theta}{\partial X^2} + \frac{\partial^2 \theta}{\partial Y^2}\right)$$

(12)

Then, the dimensionless boundary conditions for this specified problem are

$$\left. \begin{aligned}
&\text{At the vertical boundary}: U = 0, \ V = 1, \ \frac{\partial \theta}{\partial N} = 0 \\
&\text{At the horizontal boundary}: U = 0, \ V = 0, \ \theta = \sin(\pi x) \\
&\text{At the curve boundary}: U = 0, \ V = 0, \ \theta = \theta_c
\end{aligned} \right\}$$

(13)

**2.2.3 Nanofluid and its properties.** Table 1 shows the nanofluid's thermophysical properties, which are shown below, according to Roslan et al. [35]. The nanofluid properties that are used in Eqs (2)–(4) were formulated by following relations.

The nanofluids' effective density, or $\rho_{nf}$, is provided as:

$$\rho_{nf} = (1 - \phi)\rho_{bf} + \phi\rho_{sp}$$

(14)

and $\phi$ is nanofluid's volume fraction.

The Heat capacitance of the nanofluids:

$$(\rho Cp)_{nf} = (1 - \phi)(\rho Cp)_{bf} + \phi(\rho Cp)_{sp} \tag{15}$$

The dynamic viscosity:

$$\mu_{nf} = \mu_{bf}(123\phi^2 + 7.3\phi + 1) \tag{16}$$

The thermal expansion coefficient:

$$(\rho\beta)_{nf} = (1 - \phi)(\rho\beta)_{bf} + \phi(\rho\beta)_{sp} \tag{17}$$

The electrical conductivity:

$$\sigma_{nf} = \left(1 + \frac{3(\sigma_{sp} - \sigma_{bf})\phi}{\left(\sigma_{sp} + 2\sigma_{bf}\right) - (\sigma_{sp} - \sigma_{bf})\phi}\right)\sigma_{bf} \tag{18}$$

The thermal conductivity is used following the Maxwell model [36]

$$k_{nf} = k_{bf}\left(\frac{k_{sp} + 2k_{bf} + 2\phi(k_{sp} - k_{bf})}{k_{sp} + 2k_{bf} - \phi(k_{sp} - k_{bf})}\right) \tag{19}$$

Thermal diffusivity:

$$\alpha_{nf} = \frac{k_{nf}}{(\rho Cp)_{nf}} \tag{20}$$

The average Nusselt number is computed across the entirety of the arc-shaped heater and is defined as follows:

$$Nu_{avg} = -\left(\frac{1}{L}\frac{k_{nf}}{k_f}\right)\int_S \frac{\partial\theta}{\partial N}dS \tag{21}$$

The construction of streamline graphs involves the utilization of the stream function $\Psi(X, Y)$, which is defined as the non-dimensional forms $U = \frac{\partial\Psi}{\partial Y}$, $V = -\frac{\partial\Psi}{\partial X}$.

A positive value of $\Psi(X, Y)$ implies a counterclockwise rotation of the nanofluid flow, whereas a negative value of $\Psi(X, Y)$ suggests a clockwise rotation.

## 3. Numerical methodology

### 3.1 Numerical approach

Utilizing the Galerkin finite element technique, the non-dimensional Eqs (7)–(10) as well as the non-dimensional boundary conditions in Eq (11) are solved using computational software. The computational domain is discretized using this method into a limited number of elements composed of irregular triangular or rectangular cells. Finite element equations are constructed using triangular elements. The mesh structures of the full domain equipped with triangular mesh components are depicted in Fig 1. The governing nonlinear partial differential equations are then converted using the Galerkin weighted residual method to create a system of integral equations. The Gauss Quadratic Method is used to carry out the investigation required for each component of these equations. Boundary conditions are then applied to those algebraic nonlinear equations. Ultimately, the system of those nonlinear algebraic equations is solved and represented in matrices by using the Newton-Raphson Iteration method. A particular

**Table 2. Grid refinement check.**

| Mesh Elements | Boundary Elements | $Nu_{avg}$ | Time(s) |
|---|---|---|---|
| 51765 | 2565 | 1.17822563 | 34 |
| 64842 | 2600 | 1.18627288 | 38 |
| **76023** | **2729** | **1.20584875** | **45** |
| 100021 | 2975 | 1.20589929 | 55 |

upper constraint is used to set the convergence criterion and error estimation for the numerical solution. Afterwards, the process of solving the problem is carried out until the convergence requirements are met.

### 3.2 Grid test

To discretize the domain, triangular elements were used so that the computational cost could be minimized. First of all, a total of 51765 domain elements were considered, and later, the number of elements was increased step by step. The average Nu and temperature were calculated for the increased grid number. The result is depicted in Table 2. Grid sensitivity test was observed in terms of average Nu for Re = 100, Ri = 1, $\phi$ = 4%, Ha = 10, and Pr = 6.83. The analysis found that a total of 76023 is optimum number of elements to minimize the computational cost and negligible change in the result.

### 3.3 Code validation

The numerical model employed in this study has been validated against findings from previous literature. Specifically, comparisons were made between the Nusselt number, isotherm, and streamline contours to assess the accuracy of the results. Table 3 compares the average Nu on the upper surface of the triangle with the values reported by Khanafer and Chamkha [37] and Soomro et al. [38] for Gr = 100 and Pr = 0.71. The average Nusselt number has been seen to closely align with earlier investigations, exhibiting minimal error. Fig 2 depicts the comparison between Xiong et al. [39]'s study and present study in terms of isotherms(Fig 2(a)) and streamline contours (Fig 2(b)) at fixed value of Ri = 2, Re = 400 and Ha = 10 for right obstacle. It has been noted that the velocity and temperature distributions for both contours exhibit a high degree of similarity. Therefore, it can be confidently stated that the current study is well supported by prior literature. Further inquiries have been conducted in this study utilizing the identical code.

### 3.4 Artificial Neural Network analysis

An Artificial Neural Network (ANN) is a computing system that can accurately represent complex patterns and effectively handle prediction problems. This is because an ANN closely mimics the functioning of the human brain. A neural network builds a correlation between a

**Table 3. Comparison of the average Nusselt number at the top surface of the triangle with Khanafer and Chamkha [37] and Soomro et al. [38] at Gr = 100 and Pr = 0.71.**

| Parameter | Khanafer and Chamkha [37] | Soomro et al. [38] | Present study |
|---|---|---|---|
| Re = 100 | 2.01 | 2.01 | 2.03 |
| Re = 200 | 3.91 | 3.91 | 3.89 |
| Re = 400 | 6.33 | 6.33 | 6.36 |

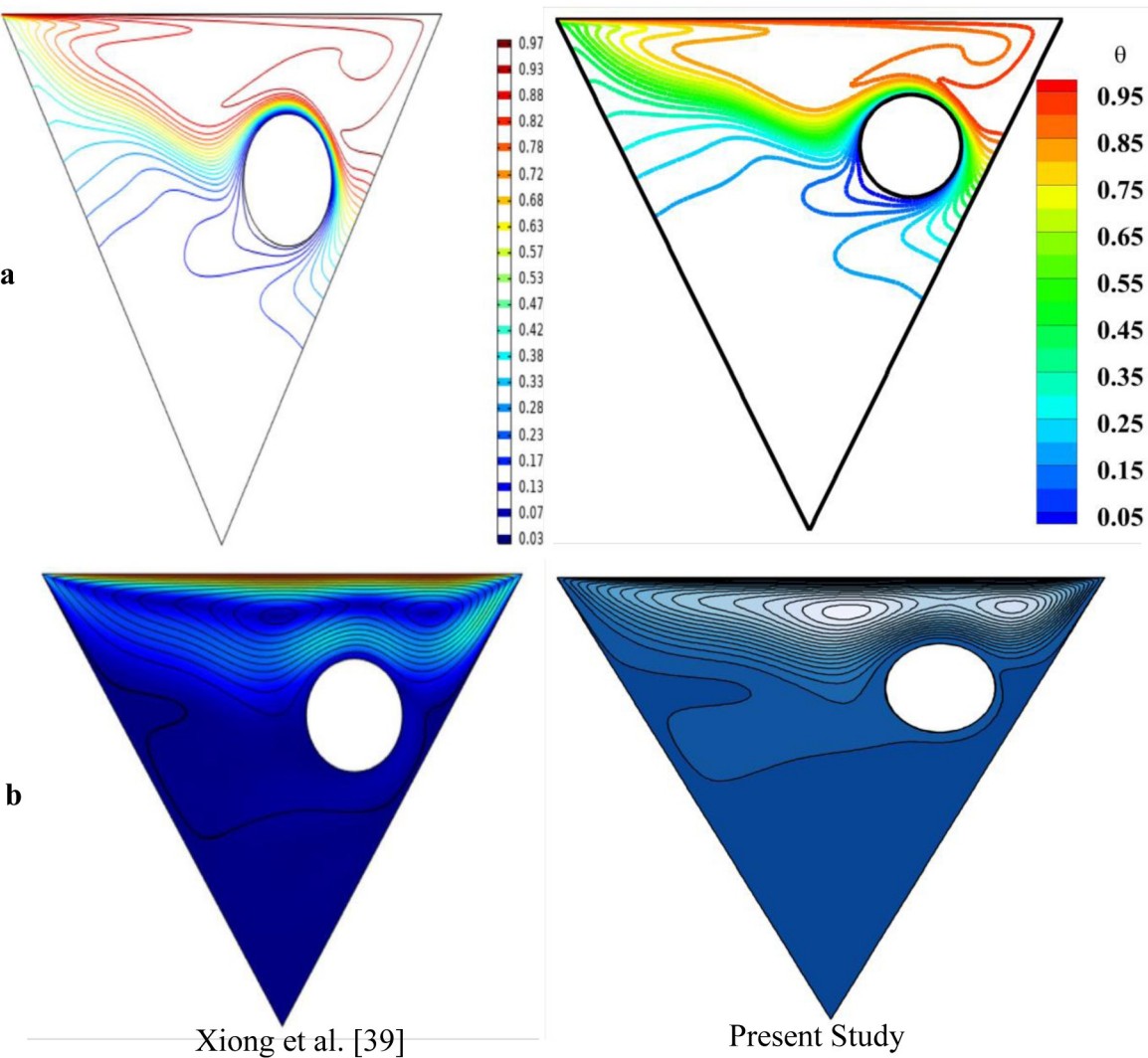

**Fig 2.** Model validation **(a)** Isotherm contours and **(b)** Streamlines contours of Xiong et al. [39] and the present study.

collection of numerical goals and a series of numerical inputs. Artificial Neural Networks (ANN) can be designed with a feed-forward, recurrent, multi-layer, or single-layer architecture. The illustrated ANN structure and Flow chart in Fig 3 comprises three layers: the input, hidden, and output layers. The chosen model for this job was a feed-forward back propagation neural network, as feed-forward networks can solve issues involving finite input-output mapping. To predict the value of the output variable ($Nu_{avg}$), a dataset consisting of 256 entries for the four input variables (Ri, Ha, $\phi$, Re) is utilized.

The process of training Neural Networks via back propagation consists of three stages: (i) the forward propagation of the input training pattern, (ii) the computation and backward propagation of the corresponding error, and (iii) the modification of the weights. This procedure can be employed with various optimization strategies. The discrepancy between the network's output and the desired value is transmitted in the opposite direction during the backward pass and utilized to modify the weights of the preceding layers. The study utilized the neural network fitting (NNF) tool available in Matlab (version R2018a) as a soft-computing

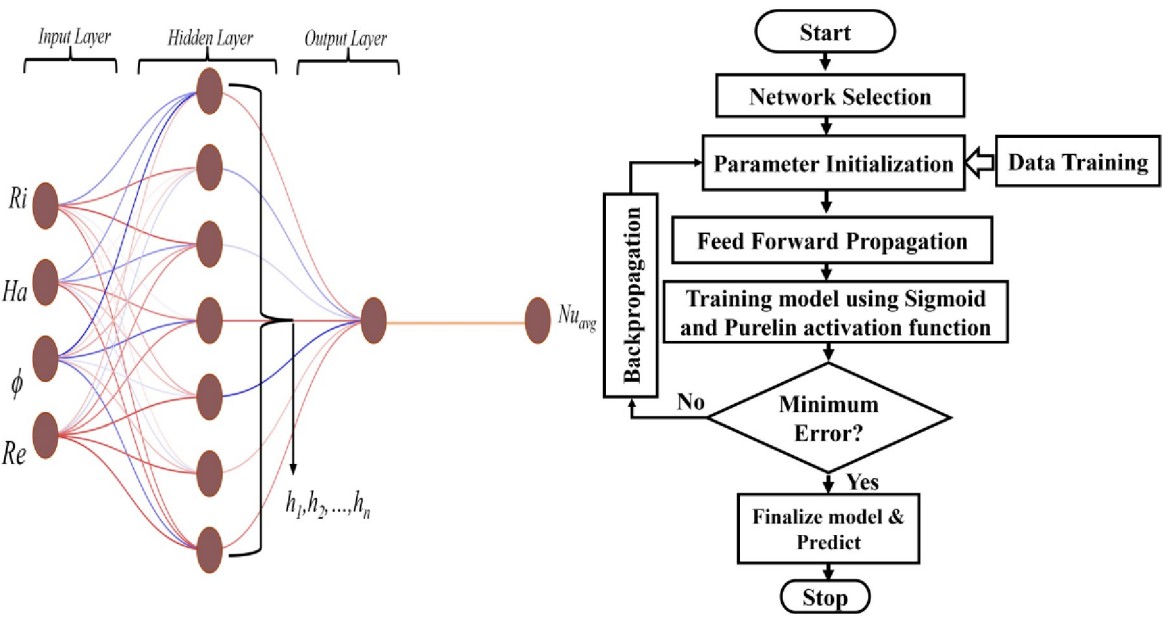

**Fig 3. ANN model diagram (left) and ANN Flow chart (right).**

tool to do neural network modeling. For fitting challenges, a neural network can be employed to establish a relationship between a dataset consisting of numerical inputs and a set of numerical goals. The NNF tool facilitates the creation, training, and evaluation of a network by utilizing mean square error and regression analysis to assess its performance.

An artificial neural network (ANN) can be converted into a mathematical equation by utilizing the weights and biases, along with the transfer function, as shown in Eq (22).

$$Y = b_0 + \sum_{k=1}^{h} \left[ w_k \times f_{sig} \left( b_{hk} + \sum_{i=1}^{m} w_{ik} X_i \right) \right] \qquad (22)$$

Then, average Nusselt number is,

$$Nu_{avg} = b_0 + \sum_{k=1}^{h} LW_k f(U_k) \qquad (23)$$

where,

$$U_k = \left( b_{hk} + \sum_{i=1}^{m} w_{ik} X_i \right) \qquad (24)$$

The equation's variables are defined as follows: $b_0$ represents the output layer's bias, $w_k$ represents the connection weight between the $k$ th neuron of the hidden layer and the single output neuron, $b_{hk}$ represents the bias at the $k$ th neuron of the hidden layer, h represents the number of neurons in the hidden layer, m represents the number of neurons in the input layer, $w_{ik}$ represents the connection weight between the ith input variable and the hidden layer, $X_i$ represents the normalized input variable $i$, $Y$ represents the normalized output variable, and $f_{sig}$ represents the transfer function.

The mathematical model proposed in this study predicts the Artificial Neural Network (ANN) using two normalized input variables (i.e. $i = 4$), one targeted output variable, seven neurons connecting the input and hidden layer (i.e. $h = 7$), and the sigmoid transfer function ($f_{sig}$ = sigmoid).

The value of $b_0$ is 6.0672705. The layer weight matrix $LW_k$ and the numerical value of neurons $U_k$ are given in Eqs (23) and (24), respectively.

$$U_k = \begin{bmatrix} 0.001826 & -1.212918 & 0.113084 & 0.648810 \\ 0.001464 & -0.457725 & 0.014228 & 0.707372 \\ 0.007624 & 9.699771 & -0.221052 & 2.287631 \\ 0.000168 & -0.048870 & -0.023647 & -0.081588 \\ 2.720902 & -0.820587 & 1.575987 & 2.920906 \\ -0.000876 & 0.686736 & -0.0397041 & -0.686933 \\ 0.004835 & -0.525021 & 0.034204 & -1.274823 \end{bmatrix}_{7 \times 4} \times \begin{bmatrix} Ri \\ Ha \\ \phi \\ Re \end{bmatrix}_{4 \times 1} + \begin{bmatrix} 2.564257 \\ 1.041381 \\ 9.045751 \\ 0.762107 \\ 1.809191 \\ 2.131071 \\ 0.948976 \end{bmatrix}_{7 \times 1} = \begin{bmatrix} U_1 \\ U_2 \\ U_3 \\ U_4 \\ U_5 \\ U_6 \\ U_7 \end{bmatrix}_{7 \times 1} \quad (25)$$

$$LW_k = \begin{bmatrix} -2.579816 \\ 1.036321 \\ 0.149146 \\ -8.650186 \\ -0.000911 \\ -4.106469 \\ 0.2984267 \end{bmatrix}_{7 \times 1} \quad (26)$$

Prior to delving into the research of ANN, it is crucial to examine the correlation between the variables. The provided information includes a Pearson correlation matrix, which is presented in Fig 4. A number approaching 1 indicates a strong positive correlation. Here, the correlation coefficient between the Nusselt number and Reynolds number is 0.96. This indicates a strong positive correlation between these variables. Conversely, the association between Reynolds number and volume fraction ($\phi$) is so small.

The schematic diagram of the ANN model is displayed in Fig 3, along with the sequential representation. There is currently no standardized approach for determining the optimal number of hidden layers. In this case, a hidden layer consisting of 7 neurons is employed to save computational time and maintain model simplicity in order to prevent unnecessary complexity.

## 4. Results and discussions

The following section discusses the effect of the Richardson number, Hartmann number, and Reynolds number varying the solid volume fraction on fluid flow, fluid temperature, and heat transfer in the triangular-shaped cavity using water-$Al_2O_3$ nanofluid.

### 4.1 Effect of Richardson number

The impact of the Richardson number on fluid velocity can be observed from the streamline plots. Fig 5 illustrates the fluid flow streamlines inside a triangular shape cavity with different mixed convection situations, varying the solid volume fraction of the nanofluid. Three Richardson number cases, Ri = 0.01, 1, and 5, have been selected, and three volume fractions of

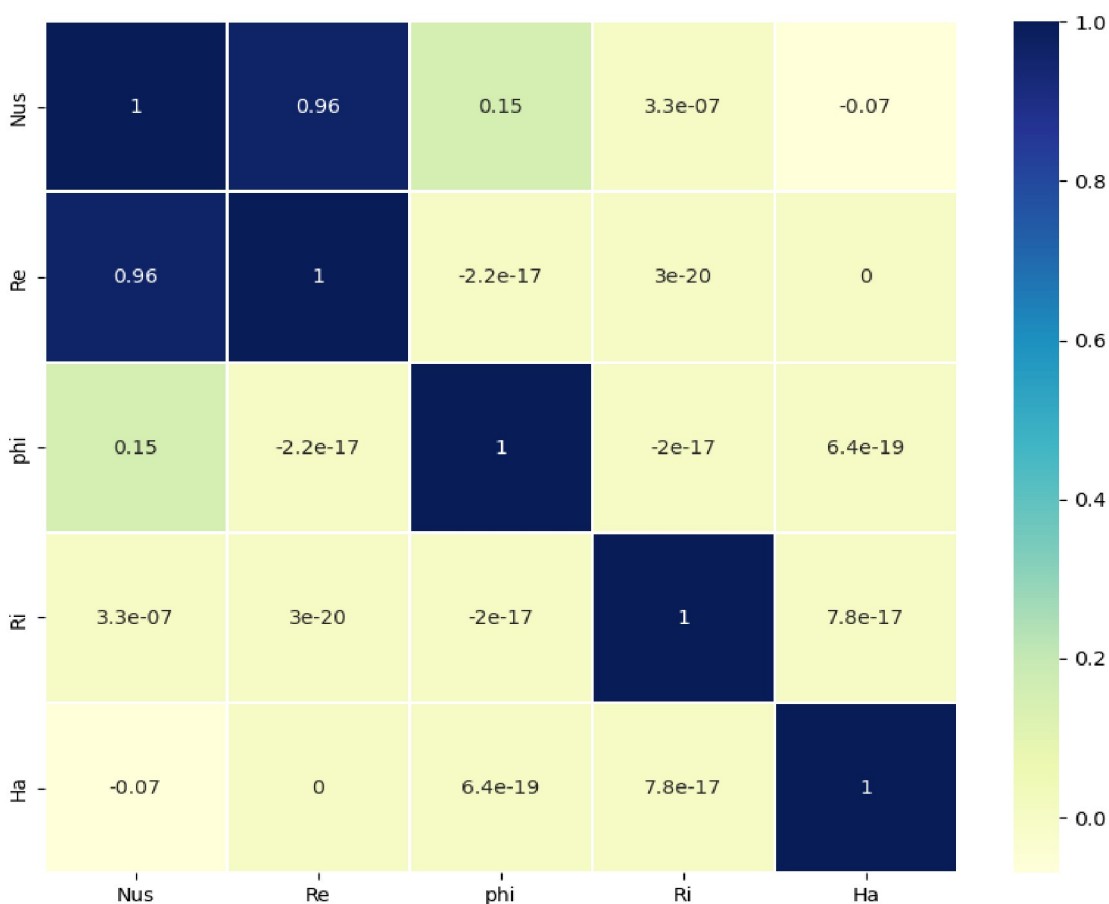

**Fig 4. Correlation of parameters with average Nusselt number.**

nanofluid, $\phi$ = 1%, 3%, and 4%, have been chosen for this analysis. Here, Ri = 0.01 represents the forced convection dominant case, and Ri = 5 denotes natural convection. From the figure, it has been observed that the fluid velocity gradient doesn't change much with the increase or decrease in the volume of the fluid. However, mixed convection plays a vital role in changing the velocity streamlines. For a specific solid volume fraction condition, the fluid velocity streamline is found to be higher for higher Richardson number cases. This is because forced convection is dominant in such cases, leading to a higher velocity gradient. Therefore, in the case of Ri = 5, the highest velocity gradient is observed. The lowest gradient is found for the natural convection dominant cases, Ri = 0.01. Considering all the conditions, the highest velocity streamline is found for the case of Ri = 5 with $\phi$ = 4%.

There is an impact on the temperature gradient inside the triangular-shaped cavity due to variations in mixed convection values and solid volume fraction. Fig 6 illustrates the isotherm plots for three variable Richardson numbers and three solid volume fractions. It is observed that with the rise of the Richardson number, the temperature gradient increases, which is obvious. This is because the higher Richardson number ensures forced convection where velocity is involved. Therefore, the temperature variation spans across the fluid domain, and the heat transfer is also higher. Higher temperatures are found for Ri = 5 cases under any solid volume fraction conditions. On the other hand, in the Ri = 0.01 cases, the temperature of the nanofluid is lower. Nanoparticle volume concentration has an impact on the temperature profile in the

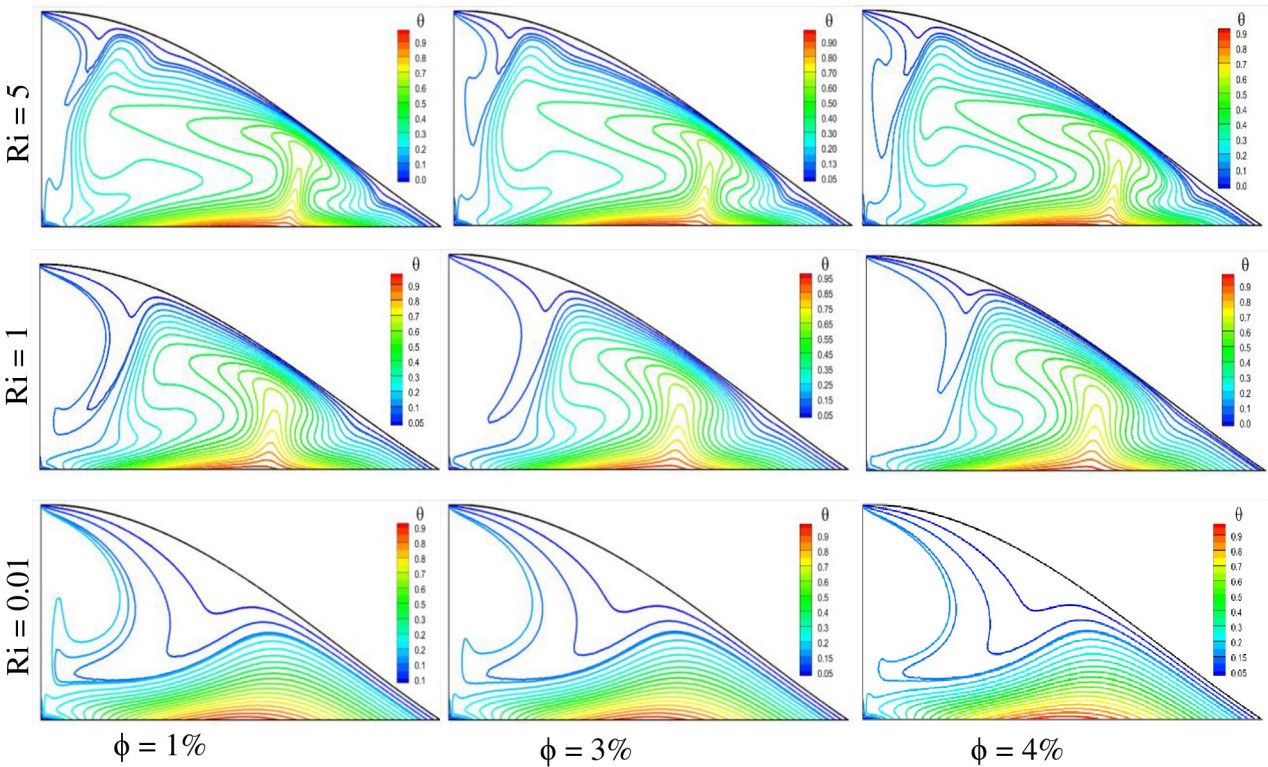

**Fig 5. Effect of Richardson number on streamlines for different volume fractions at Pr = 6.83, Re = 100, and Ha = 10.**

fluid domain. The temperature of the fluid rises with the increase of nanoparticle concentrations. This is because the thermal conductivity is higher for a more concentrated nanofluid. As a result, the higher temperature is found for $\phi$ = 4%. Comparing the nine cases, it is noticed that the highest temperature is found for the case of Ri = 5 with $\phi$ = 4%, and the lowest temperature is obtained for Ri = 0.01 with $\phi$ = 1%.

The rate of heat transmission is contingent upon the regime of mixed convection and the concentration of nanofluid. The Nu represents the rate at which heat is transferred. Fig 7 depicts how heat is transferred for different Ri and volume concentration scenarios. Fig 7(a) displays the line graph representing the Nu for a range of Ri values from 0.01 to 5 and $\phi$ values from 1% to 4%. It has been noted that the Nu escalates as the concentration of nanofluid increases. This is because the higher concentration of nanofluid leads to an increase in thermal conductivity. Consequently, the heat transfer rate is larger when $\phi$ equals 4%, and the Nu value is lower when $\phi$ equals 1%. The heat transfer rate also depends on the value of the Ri. Fig 7(b) displays a two-dimensional graph illustrating the Nu as it varies with different values of Ri and $\phi$. It has been observed that the rate of heat transmission increases as the Ri increases. During forced convection, there is an increase in the rate of heat transfer, which is a common occurrence. Additionally, it is seen that the heat transfer rate reaches a steady state at a Ri of 4. Furthermore, increased forced convection leads to a decrease in the average Nu. Fig 7(c) displays a 3D figure, revealing that the maximum Nu is seen for the scenario with Ri = 4 and $\phi$ = 4%. The situation with Ri = 0 and $\phi$ = 1% yields the minimum heat transfer rate.

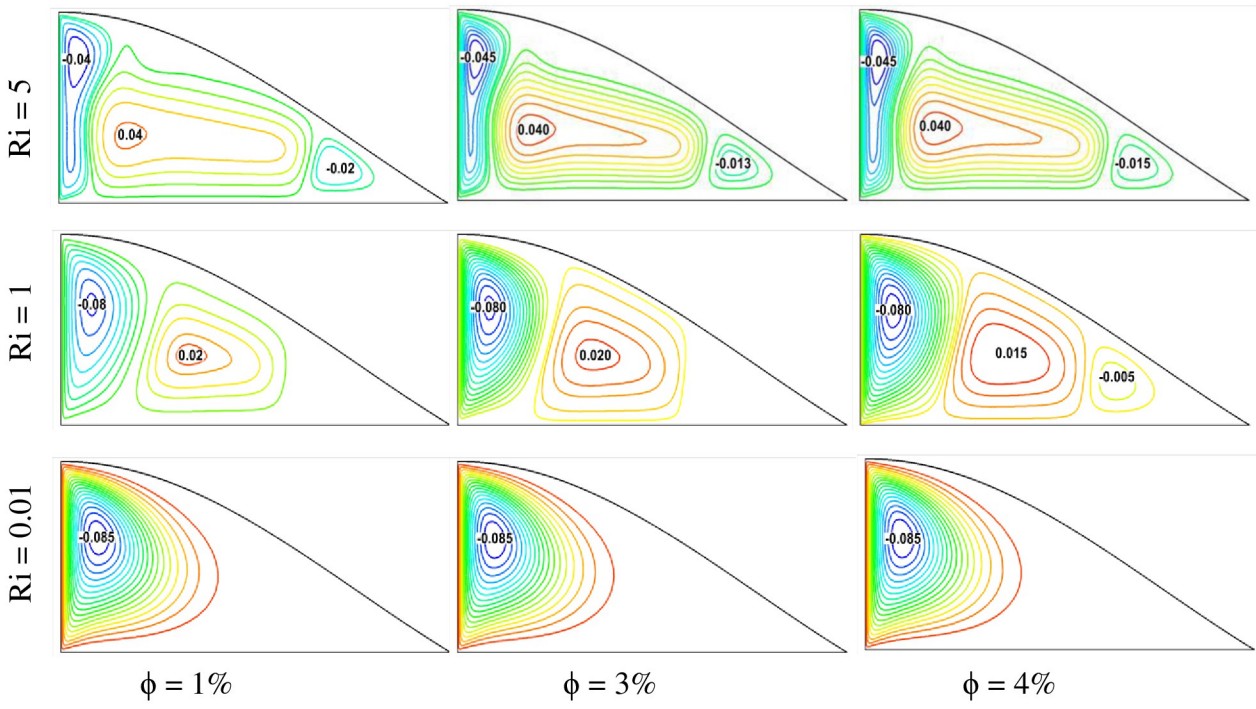

**Fig 6. Effect of Richardson number on isotherms with different volume fractions at Pr = 6.83, Re = 100, and Ha = 10.**

## 4.2 Effect of Reynolds number

This study also analyzes fluid flow, temperature, and heat transfer, considering variable fluid velocity. Fig 8 shows the velocity streamline plots for different Reynolds numbers and solid volume fractions. Three different Reynolds numbers, Re = 50, 100, and 200, were chosen, and three variable solid volume fractions ϕ = 1%, 3%, and 4% were selected for the study. It is observed that the fluid velocity gradient increases with the rise of the solid volume fraction. This happens for all Reynolds number conditions. For a specific Reynolds number, with the augmentation of nanoparticle concentration, the velocity gradient is found to be higher. As a result, higher velocity is found for ϕ = 4%. Reynolds number plays a vital role in augmenting the velocity gradient in the triangular-shaped cavity. It is found that the higher the Reynolds number, the greater the velocity streamline. This is because a higher Reynolds number denotes higher velocity. As the higher velocity is involved for the greater Reynolds number, the velocity streamlines are found to be the highest in the case of Re = 200 for any constant solid volume concentration cases. On the other hand, the velocity streamline is found to be lesser for the lower Re case, such as Re = 50. Comparing all the cases, the greater velocity streamline is found for the case of Re = 200 with ϕ = 4%, and the lowest velocity profile is obtained for Re = 50 with ϕ = 1%.

The temperature profile of the triangular-shaped cavity for variable Reynolds numbers can be observed from the isotherm plots shown in Fig 9. The temperature of the fluid is not the same for different Reynolds numbers and solid volume fraction cases. It is observed from the plots that the Reynolds number has a positive effect on the fluid temperature. The higher the Reynolds number, the greater the temperature of the fluid is found. This is because higher fluid velocity ensures better heat dissipation inside the cavity, resulting in higher fluid temperature. For a constant solid volume fraction case, the higher fluid temperature is found for

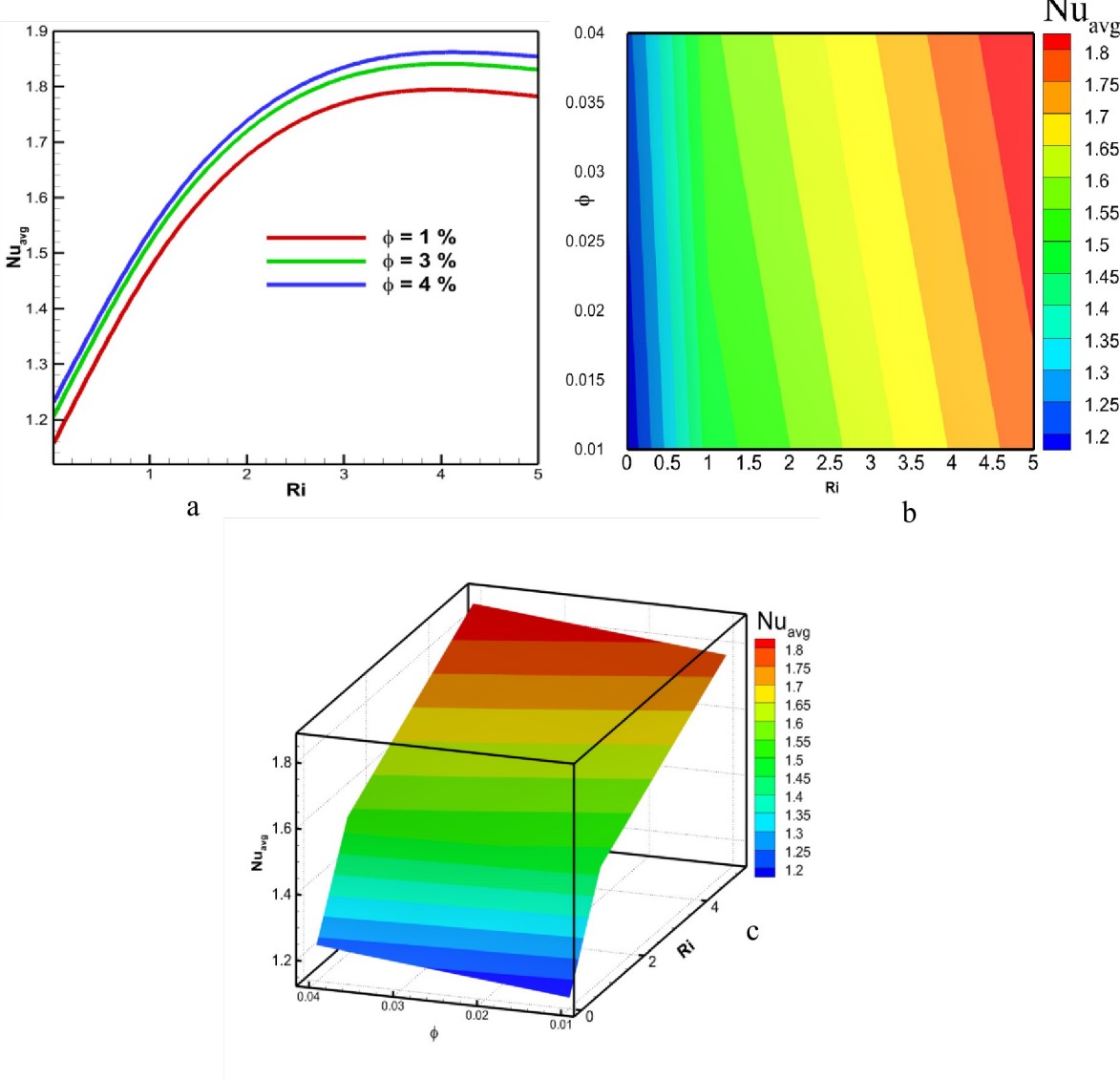

**Fig 7. Effect of Richardson number and volume fraction on average Nusselt number at Pr = 6.83, Re = 100, and Ha = 10.**

Re = 200, whereas the lower is obtained for Re = 50. The fluid temperature also rises with the solid volume fraction as before. The cases with $\phi$ = 4% show higher fluid temperature, and $\phi$ = 1% show lower temperature. As a result, the highest fluid temperature is found for the case of Re = 200 with $\phi$ = 4%, and the lowest fluid temperature is reported for Re = 50 with $\phi$ = 1%.

The heat transfer rate depends on the Reynolds number and solid volume fraction as well. The variation of heat transfer rate for changed values of Reynolds number and volume fraction has been shown in Fig 10. It is found that the heat transfer rate increases with the increase in fluid velocity. A higher Reynolds number ensures a better heat transfer rate, and therefore, a higher Nusselt number is found for the case of Re = 400. However, the lower Nusselt number is reported for lower Reynolds numbers, such as Re = 100. It is also noticed that the higher transfer rate rises very little with the change in fluid velocity. On the other hand, solid volume

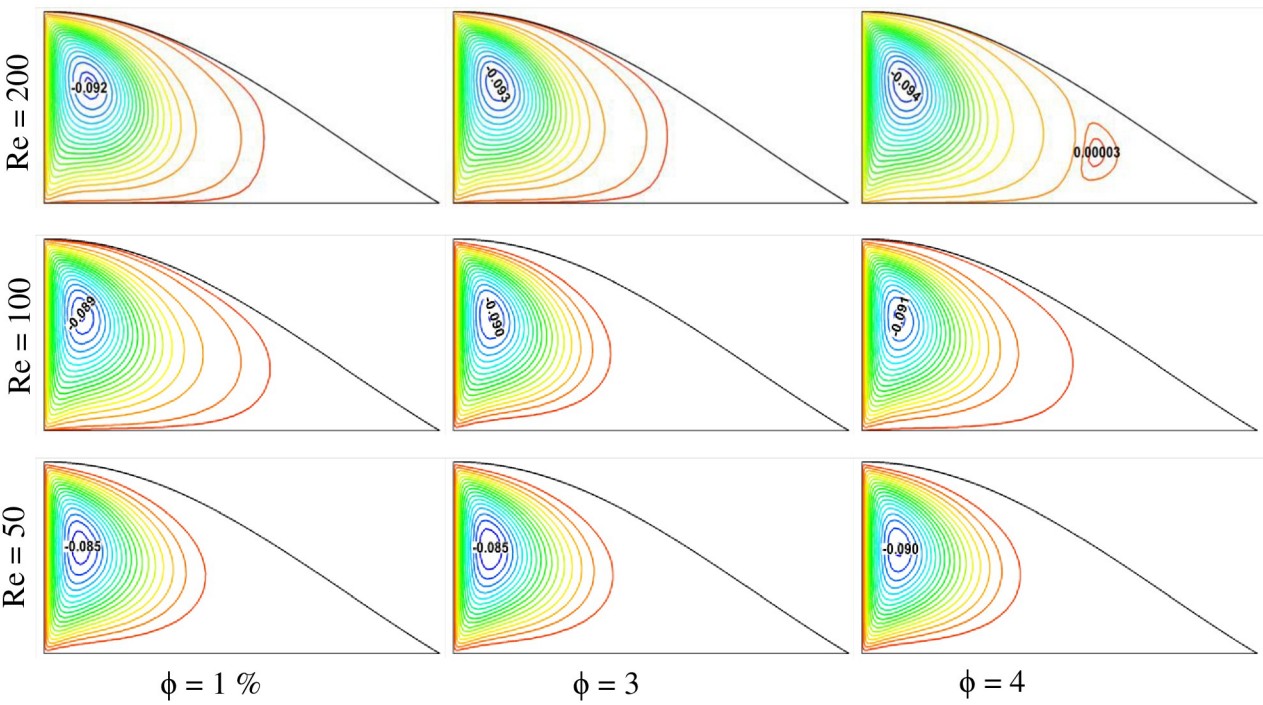

**Fig 8. Effect of Reynolds number on streamlines for different volume fractions at Pr = 6.83, Ri = 1, and Ha = 10.**

fraction plays a more vital role than the Reynolds number. The heat transfer rate increases for higher values of solid volume fraction. As a result, the highest heat transfer rate is found for the case of $\phi = 4\%$, as before. The reason is that the more concentrated nanofluid is highly capable of transporting heat as the thermal conductivity is greater. However, the lower heat transfer rate is observed for the lower nanofluid concentration $\phi = 1\%$ due to lower thermal conductivity. The highest Nusselt number is found for the case of Re = 400 with $\phi = 4\%$. On the other hand, the lowest heat transfer rate is obtained for Re = 100 with $\phi = 1\%$.

### 4.3 Effect of Hartmann number

The impact of magnetohydrodynamics was also analyzed by varying the Hartmann number in the triangular-shaped cavity. Fig 11 represents the streamlines plot in the cavity for different Hartmann numbers (Ha = 0, 20, and 100) and three variable volume fractions $\phi = 1\%$, 3%, and 4%. It is observed that the fluid flow streamline has a negative effect on the Hartmann number. With the increase of the magnetic field, the velocity streamline gets reduced. This is because the fluid flow has been hindered due to magnetohydrodynamics. However, the velocity profile enlarges for higher values of solid volume fraction. For the Ha = 0 and 20 cases, a separate vortex region is also found on the right side of the cavity for the higher volume fraction case $\phi = 4\%$. The fluid velocity gradient is higher for Ha = 0, and the lowest is obtained for Ha = 100. Regarding the nanofluid concentration, a higher velocity profile is seen for $\phi = 4$, while the lower one is observed for $\phi = 1\%$.

The magnetic effect also has a negative impact on the temperature of the fluid. This is evident from the isotherm plots in Fig 12, where three Hartmann numbers and three volume fraction cases are presented. It is observed that, due to the increase in the Hartmann number, the temperature gradient is reduced. Conversely, it rises for lower values of the Hartmann

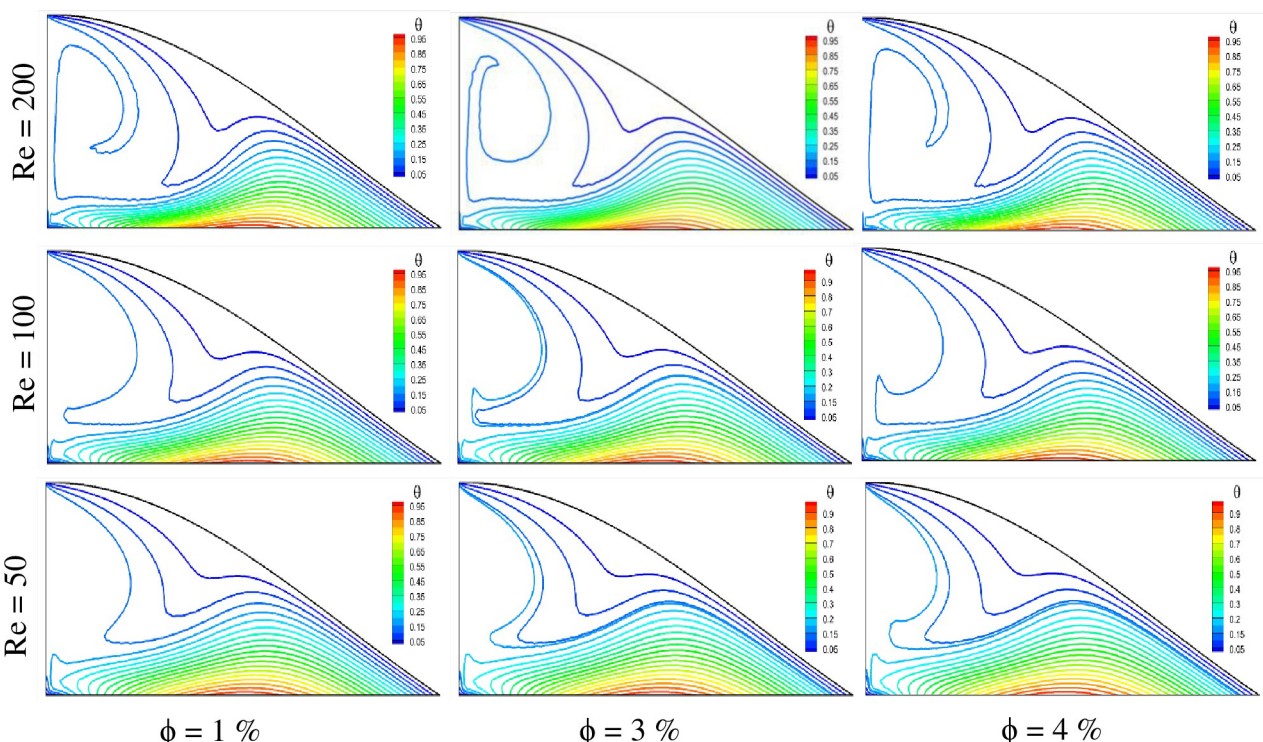

**Fig 9. Effect of Reynolds number on isotherms for different volume fractions at Pr = 6.83, Ri = 1, and Ha = 10.**

number, where the magnetic field is lower. Consequently, the highest temperature gradient is found for the Ha = 0 case, and the lowest temperature gradient is observed for Ha = 100 in a constant volume fraction case. The solid volume fraction contributes to an increase in fluid temperature, with the fluid temperature rising as the volume fraction increases. Therefore, the highest temperature is observed for $\phi$ = 4%, and the lowest is obtained for $\phi$ = 1%. Comparing all the cases, the highest fluid temperature is found for the case of Ha = 0 with $\phi$ = 4%, and the lowest is observed for Ha = 100 with $\phi$ = 1%.

Fig 13(a)–13(c) illustrate the comparison of the Nusselt number for different Hartmann numbers and solid volume fractions in the triangular-shaped cavity using line graph, 2D plot and 3D plot. It is observed that the heat transfer rate decreases with the increase in Ha. This is because the Hartmann number has a negative effect on fluid flow and heat transfer. The heat transfer rate significantly reduces from Ha 100 to 350. Beyond 350, the rate of heat transfer decrement is lower up to 400. Consequently, the lowest heat transfer rate is found for Ha = 400. A similar trend is observed for all solid volume fraction cases. However, for an increased value of solid volume fraction, the Nusselt number rises. The highest Nusselt number is found for $\phi$ = 4%, and the lowest is obtained for $\phi$ = 1%. This is attributed to the higher thermal conductivity capable of high heat transfer for high-concentration nanofluids. Comparing all cases, the highest Nusselt number is found for Ha = 100 with $\phi$ = 4%, and the lowest is obtained for Ha = 400 with $\phi$ = 1%.

## 4.4. ANN training, testing and validation

In this study, the average Nusselt number ($Nu_{pred}$) is estimated using a two-layer feed-forward network. The accuracy of the estimation is verified by calculating the Mean Squared Error

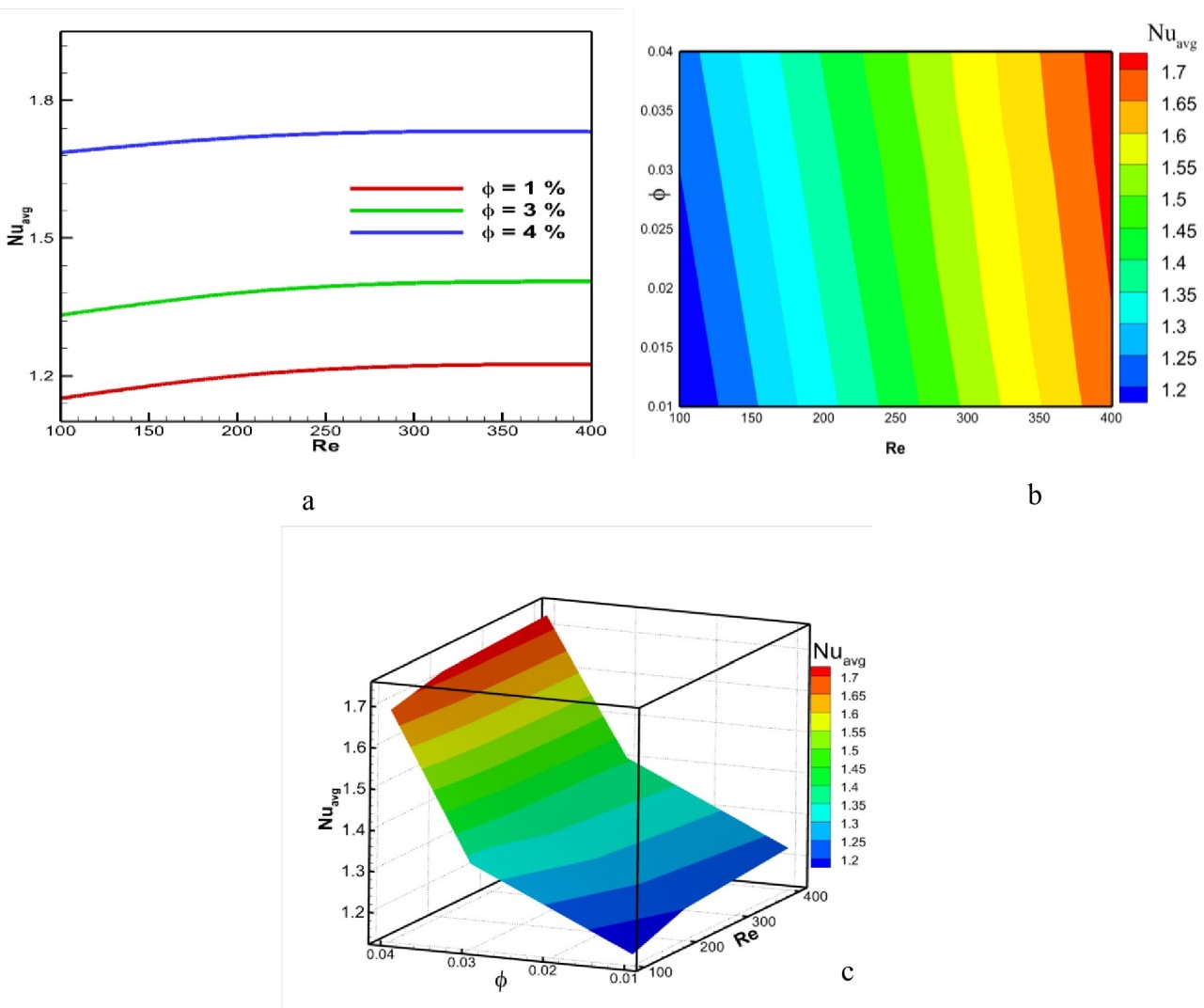

**Fig 10. Effect of Reynolds number and volume fraction on average Nusselt number at Pr = 6.83, Ri = 1, and Ha = 10.**

(MSE). The Levenberg-Marquardt backpropagation algorithm is used to train the suggested network. In this work, the hidden layer utilized the sigmoid activation function as its transfer function, whereas the output layer employed either a linear transfer function or the purelin activation function. For this ANN study, phi (0.01,0.02,0.03,0.04), Ha (0,20,50,100), Re (100,200,300,400) and Ri (0.01,0.1,1,5), these combinations were taken. And 70% of data i.e., 180 observations, were taken to feed the model, 15% of data i.e., 38 obs. To ensure validation, an additional 15% of the data is allocated to validate the accuracy of the model. The model's accuracy is represented in Table 4, which displays the Mean Squared Error (MSE) value and the regression (R) value.

Table 4 clearly indicates that the Mean Squared Error (MSE) value for the training data is 5.6e-7 and for the validation data, it is 1.05e-6. These values demonstrate that the model exhibited a high level of effectiveness in predicting the target variable. The regression values for the train, validation, and test data are 0.999994, 0.999988, and 0.999989, respectively. Table 5 displays the model's effectiveness in terms of Mean Squared Error (MSE) and R for randomly

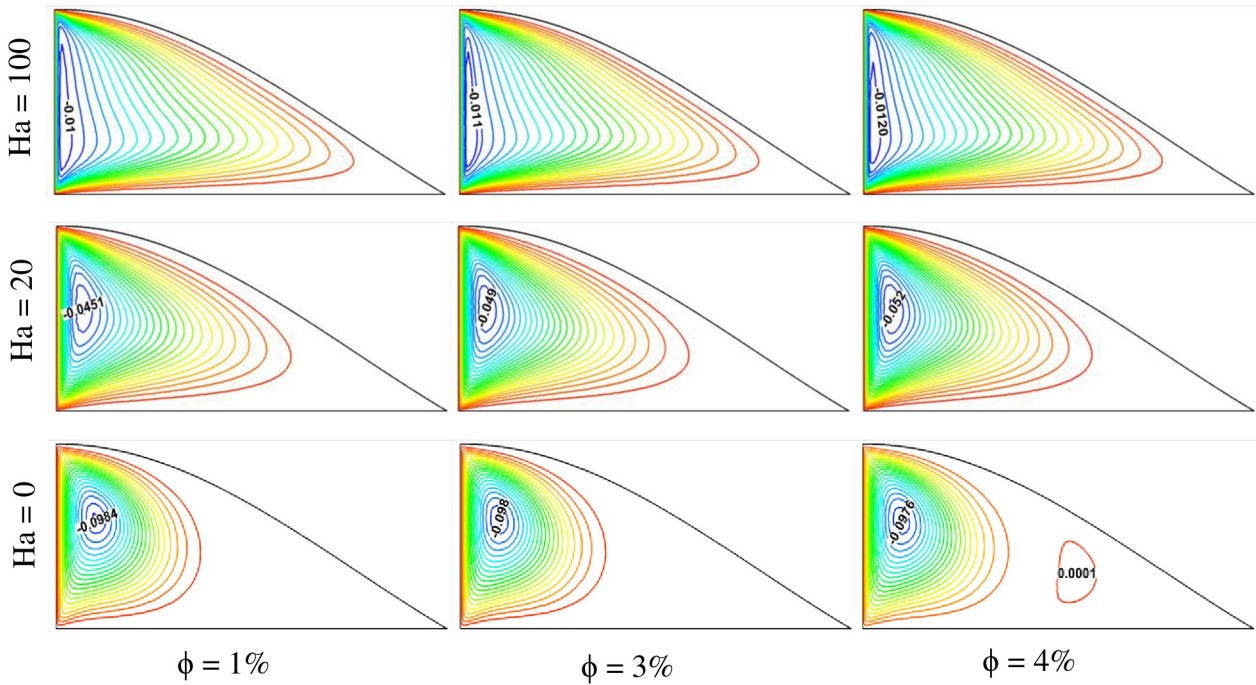

**Fig 11. Effect of Hartmann number on streamlines for different volume fractions at Pr = 6.83, Ri = 1, and Re = 100.**

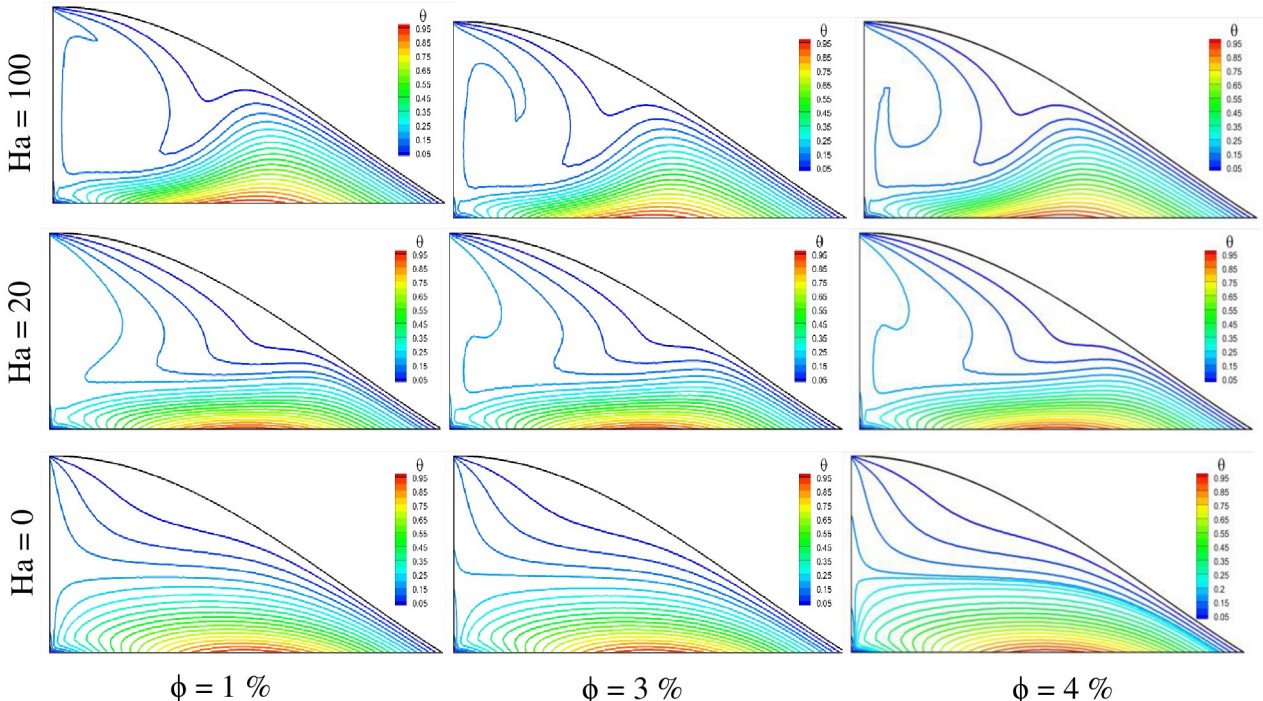

**Fig 12. Effect of Hartmann number on isotherms for different volume fractions at Pr = 6.83, Ri = 1, and Re = 100.**

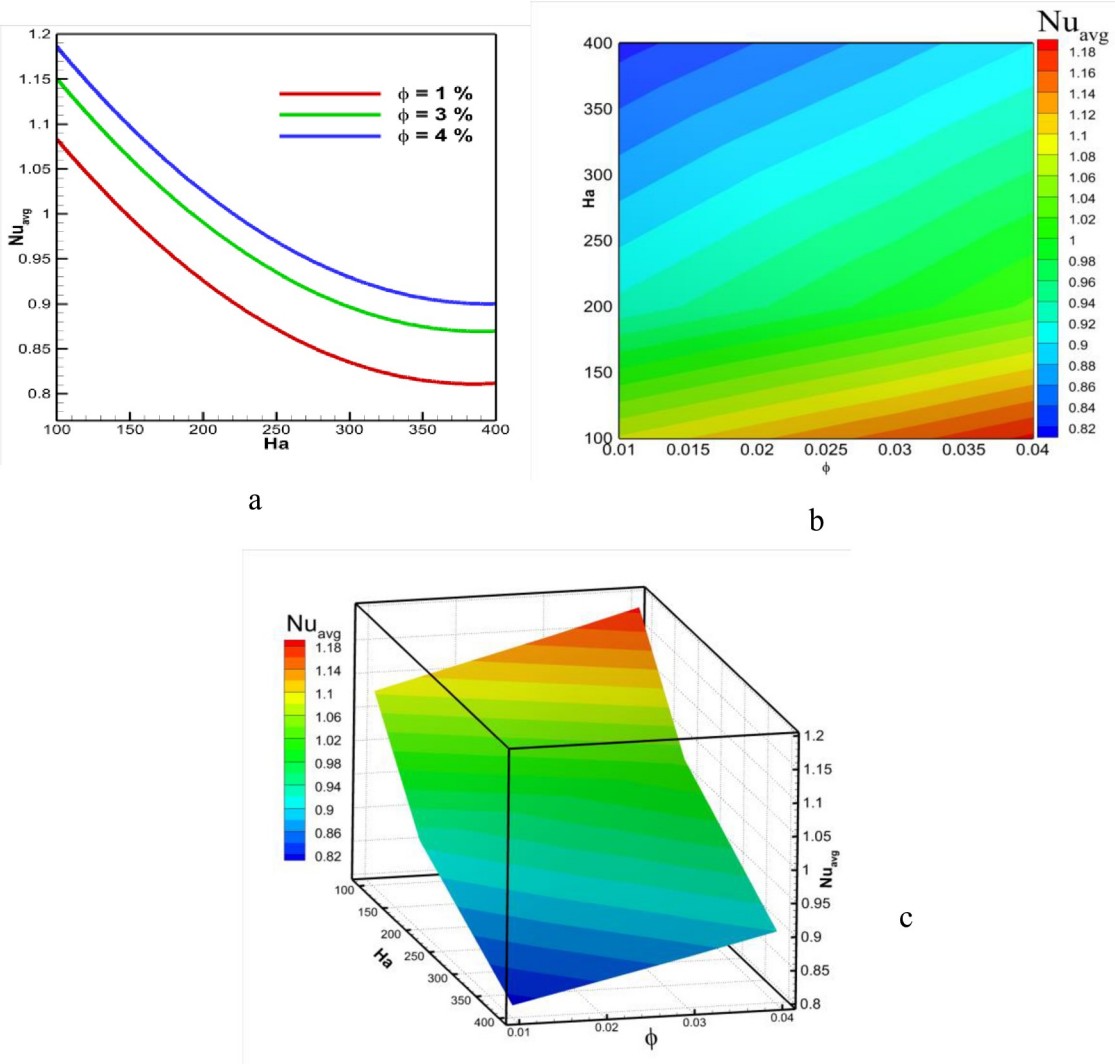

**Fig 13. Effect of Hartmann number and volume fraction on average Nusselt number at Pr = 6.83, Ri = 1, and Re = 100.**

sampled data. This further substantiates the accuracy of the model in predicting the target variable, where Regression values quantify the degree of correlation between the outputs and objectives. An R value of 1 indicates a strong correlation, while a value of 0 suggests no correlation. The MSE (Mean Squared Error) is the average of the squared differences between the predicted outputs and the actual goals. Smaller MSE values indicate better accuracy, and a value

**Table 4. MSE and R score for train (70%), Test (15%) and validation (15%) dataset.**

|  | Samples | MSE | R |
|---|---|---|---|
| Train data | 180 | 5.62841e-7 | 0.999994 |
| Validation data | 38 | 1.05200e-6 | 0.999988 |
| Test data | 38 | 7.74994e-7 | 0.999989 |

**Table 5. Model evaluation accuracy using sample data.**

| MSE | 6.66942e-7 |
|-----|------------|
| R | 9.99993e-1 |

of zero signifies no error.

$$MSE = \frac{1}{n}\sum_{i=1}^{n}\left(Nu_{avg} - Nu_{pred}\right)^2 \tag{27}$$

$$Purelin: \ f(x) = x \tag{28}$$

$$Sigmoid: \ f(x) = \frac{1}{1 + e^{-x}} \tag{29}$$

$$Error = Nu_{avg} - Nu_{pred} \tag{30}$$

Here, $Nu_{pred}$ stands for predicted Nusselt number using ANN model, $Nu_{avg}$ for simulated Nusselt number using mathematical simulation toolbox (input data), and n for data size or number of observations.

Fig 14 describes the activation function Sigmoid and purelin graphically. The activation functions that were employed in the hidden layers are found in Eqs (26) and (27), while the MSE Eq (25). The purelin activation function, also known as the linear activation function, is used in specific scenarios within ANNs. Purelin primarily shines in the output layer of ANNs designed for regression tasks. In regression, the network aims to predict continuous values, line house prices pr stock values, in this case it is average Nusselt number that is a continuous variable. Purelin, in contrast to other activation functions, outputs the weighted sum of its inputs exactly as it is. This indicates that the final predictions from the network stay within the original range of your target values, which is crucial for accurate regression. On other hand, the sigmoid activation function, also known as the logistic function, has its place in different scenarios within ANNs. Sigmoid excels in tasks where the output needs to be a probability between 0 and 1. It is widely used in hidden layers of shallow neural networks (few layers) due to its simplicity and ease of calculation.

Fig 15 illustrates a histogram of errors, and it is visible from the graph that those errors are so small in terms of number. And most of them are close to zero. In this figure, it describes a comparison for forecasted error value for Training, Test and Validation dataset. The zero line is shown at the histogram's central point. Furthermore, it is evident that the model worked incredibly well and that the predicted values were extremely precise because the difference between the targets and outputs was so little, as the error values were almost exactly zero. Additionally, the maximum instances maintain a close proximity to the zero-error line, indicating an optimal model.

Fig 16 shows the regression plot between given target value and predicted output value for three dataset such as train, test and validation. The regression line displays exact accuracy in the training case when comparing the predicted values (output) to the known values (target). It holds true in other cases as well. Additionally, a regression value that is as near to 1 suggests a reliable forecasting model. Furthermore, the regression plot predicts the desired variables and displays encouraging findings.

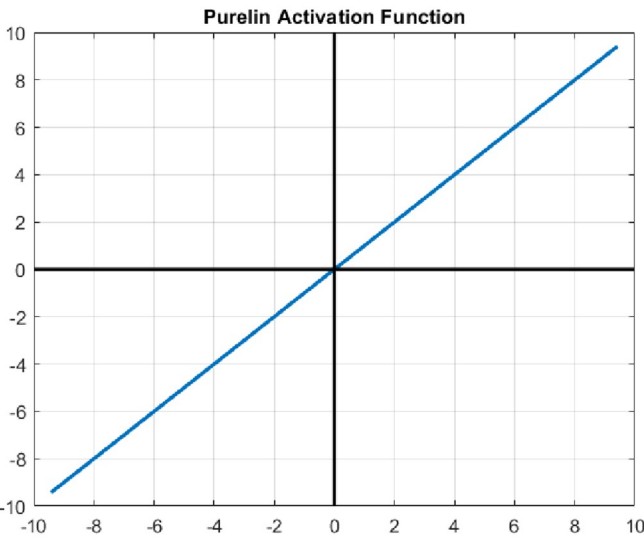

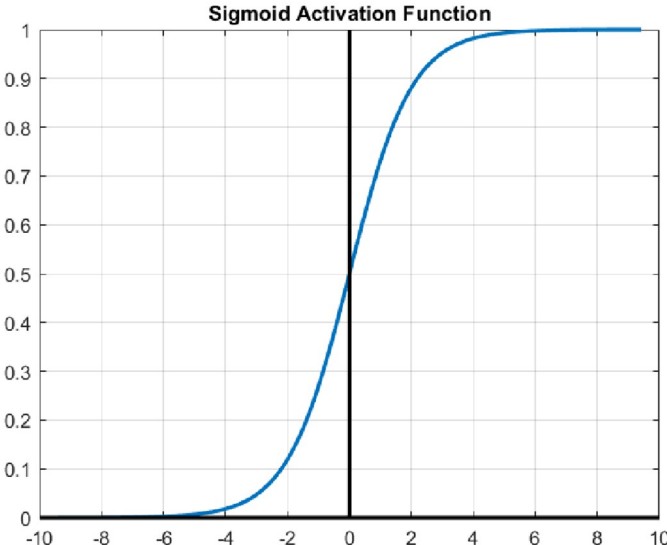

**Fig 14. Activation function (sigmoid and purelin).**

The predicted data and the Error of the predicted data are displayed in Table 6. Here, $Nu_{avg}$ is the simulated or given Average Nusselt Number. Additionally, average Nusselt number enhancement is given in the table for different values. The results showed that the Reynolds number (Re) had the greatest influence on the Nusselt number, with an increase of 17% when Re increased from 300 to 400. The influence of the Rayleigh number (Ra) was minimal, while the Ha had a negative impact, initially it increased but then decreasing the Nusselt number as Ha increased. The solid volume fraction also had a positive influence on the Nusselt number, increasing the heat transfer rate by almost 7% as solid volume fraction increases from 1% to 4%. The error was calculated using Eq (28). And it is crystal clear that the error is so minimal.

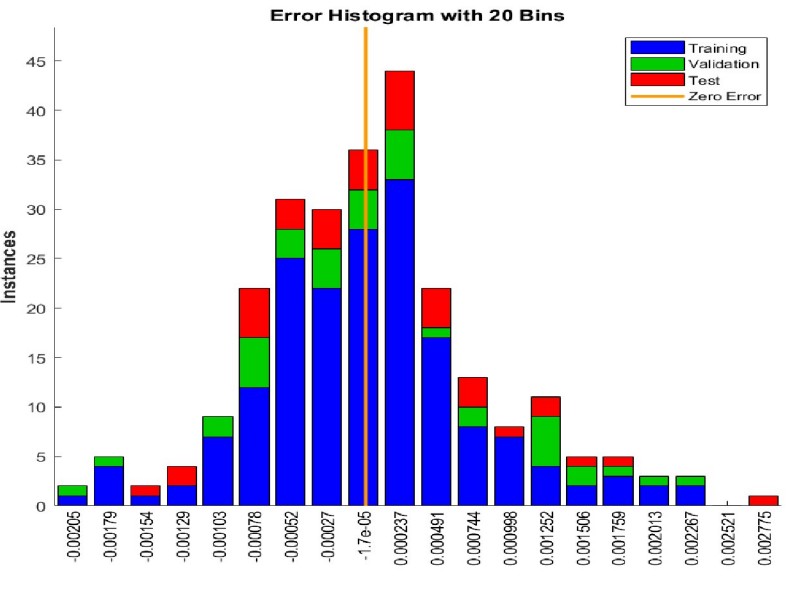

**Fig 15. Error histogram of train, test and validation dataset.**

## 5. Conclusions

This investigation was conducted to study the fluid flow and thermal performance in a triangular-shaped cavity using a nanofluid consisting of $Al_2O_3$ particles suspended in water. The study considered the effects of mixed convection, magnetohydrodynamics (MHD), and varying Reynolds numbers. Various values of the Richardson number (Ri), Hartmann number (Ha), Reynolds number (Re), and solid volume percentage of the nanofluid ($\phi$) were examined. The Levenberg-Marquardt backpropagation approach was employed to do an analysis utilizing Artificial Neural Network (ANN).

The primary discoveries of the investigation are as follows:

- The concentration of nanofluid has a major impact on the rate of heat transfer, with higher concentrations of nanoparticles leading to improved heat transfer. A concentration of 4% shown superior thermal performance in all cases.

- In cases when forced convection is the dominant factor, there is an increase in thermal transfer within the cavity. The Nusselt number reached its maximum value of 1.85 at Ri = 4, whilst the heat transfer rate was at its minimum when Ri = 1 with the value of 1.72. Furthermore, greater Richardson number values resulted in elevated temperature gradients.

- The rate of heat transmission is directly influenced by the velocity of the fluid. The Nusselt number reached its maximum value of 1.7 at a Reynolds number of 400 and a concentration of 4%, whilst the minimum heat transfer rate (Nu = 1.6) was observed at a Reynolds number of 100. Increased velocities were correlated with larger streamline velocity and temperature gradients.

- Magnetohydrodynamics (MHD) has a detrimental effect on the transmission of heat and the movement of fluid in the cavity. Higher Hartmann numbers lead to reduced fluid flow, decreased temperature gradients, and lower Nusselt numbers. The Nusselt number reached

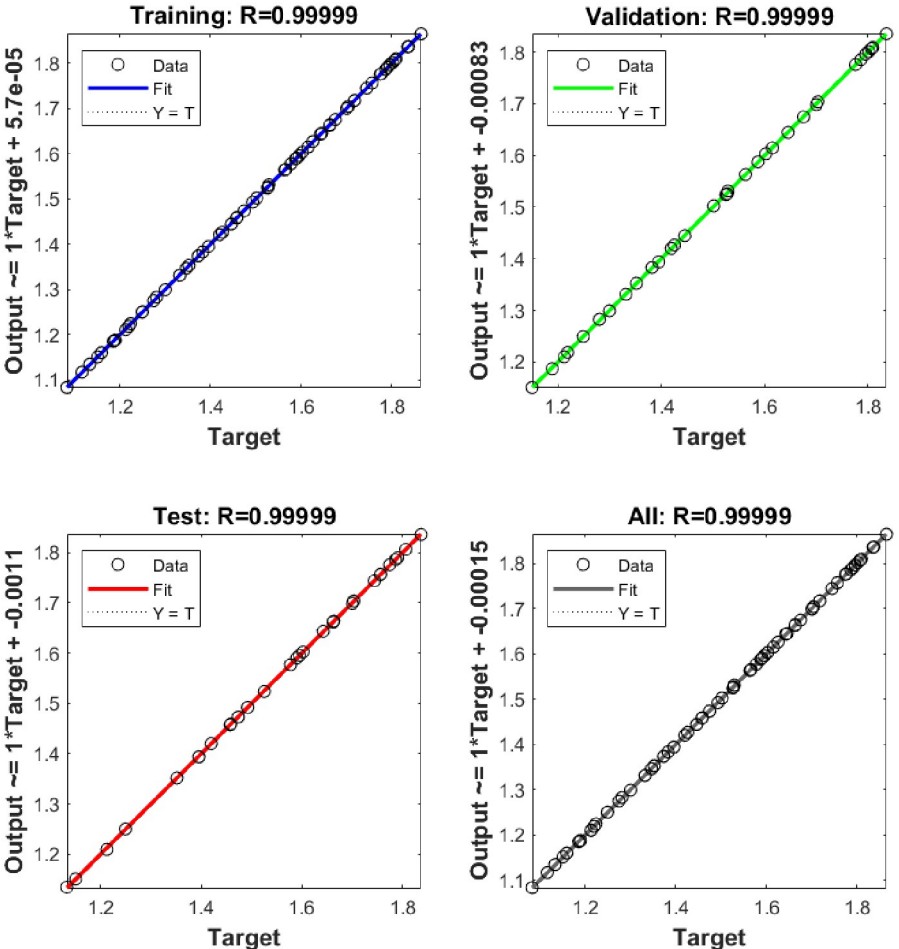

**Fig 16. Regression plot of input data (target) vs predicted data (output) of train, test and validation dataset.**

its maximum value of 1.19 at Ha = 100, but its minimum value (0.9) was observed at Ha = 400.

- The analysis using Artificial Neural Networks (ANN) demonstrated exceptional accuracy, as seen by the low Mean Squared Error (MSE = 1.05200e-6) and high R values (0.999988) for the validation data. The low error rate further confirms the efficacy of ANN in accurately forecasting the heat transfer performance of the triangular-shaped cavity under study.

This study helps fill the research gap by offering valuable insights into the fluid flow and thermal performance within triangular-shaped cavities, utilizing nanofluids and employing artificial neural network (ANN) analysis. This study improves our understanding of improving the thermal behavior of systems by examining the effects of nanofluid concentration, forced convection, fluid velocity, and MHD.

The practical applications of these results have far-reaching ramifications for the design and optimization of thermal devices and systems. The findings of this study can be utilized to increase the efficiency of heat transmission and boost the performance of triangular-shaped cavities in different engineering applications.

**Table 6. Comparison between given average Nusselt number (Nu$_{avg}$) and predicted Nusselt number (Nu$_{pred}$).**

| Ri | Ha | φ | Re | Nu$_{avg}$ | Increased (%) | Nu$_{pred}$ | Error |
|---|---|---|---|---|---|---|---|
| 0.01 | 0 | 0.01 | 100 | 1.13358920 | - | 1.13476073 | -0.00117 |
| 0.1 | | | | 1.13358920 | 0 | 1.13476276 | -0.00117 |
| 1 | | | | 1.13358921 | 8.8215e-07 | 1.13478268 | -0.00119 |
| 5 | | | | 1.13358922 | 1.7643e-06 | 1.13427070 | -0.00068 |
| 1 | 0 | 0.01 | 100 | 1.13358921 | - | 1.13478268 | -0.00119 |
| | 20 | | | 1.22359022 | 7.93947307 | 1.22427041 | -0.00068 |
| | 50 | | | 1.18927565 | 4.91240032 | 1.18743578 | 0.00184 |
| | 100 | | | 1.08327216 | -4.4387375 | 1.08320838 | 0.000064 |
| 1 | 0 | 0.01 | 100 | 1.13358921 | - | 1.13478268 | -0.00119 |
| | | 0.02 | | 1.15935672 | 2.27309062 | 1.16008965 | -0.00073 |
| | | 0.03 | | 1.18570078 | 4.59704182 | 1.18534458 | 0.000356 |
| | | 0.04 | | 1.21269731 | 6.9785509 | 1.21031513 | 0.002382 |
| 1 | 0 | 0.01 | 100 | 1.13358921 | - | 1.13478268 | -0.00119 |
| | | | 200 | 1.33156878 | 17.4648425 | 1.33132400 | 0.000245 |
| | | | 300 | 1.56502207 | 38.0590126 | 1.56467947 | 0.000343 |
| | | | 400 | 1.77549948 | 56.6263567 | 1.77553752 | -3.8e-05 |

To enhance our comprehension of fluid flow and thermal performance, it is advisable to investigate alternative nanofluid compositions and cavity configurations in future studies. Furthermore, conducting research on the impacts of other boundary conditions and analyzing more intricate geometrical configurations will yield significant knowledge. Moreover, investigating sophisticated artificial neural network structures and optimization techniques might improve the precision and effectiveness of ANN analysis for comparable systems.

## Supporting information

**S1 Table. Forecasted average Nusselt number (Nu$_{pred}$) and given average Nusselt number (Nu$_{avg}$).**
(DOCX)

## Author Contributions

**Conceptualization:** M. M. Rahman.

**Formal analysis:** M. N. Hudha, Md. Jahid Hasan, A. K. Azad.

**Investigation:** M. N. Hudha, Md. Jahid Hasan, T. Bairagi, A. K. Azad.

**Methodology:** T. Bairagi, A. K. Azad.

**Supervision:** M. M. Rahman.

**Validation:** T. Bairagi.

**Writing – original draft:** M. N. Hudha, Md. Jahid Hasan.

**Writing – review & editing:** A. K. Azad, M. M. Rahman.

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
