## [Decision Letter · Decision Letter 0]

27 Mar 2024

PONE-D-24-08432Artificial Neural Network Analysis on the Effect of Mixed Convection in a Triangular-shaped Geometry using Water-based Al2O3 NanofluidPLOS ONE

Dear Dr. Rahman,

Thank you for submitting your manuscript to PLOS ONE. After careful consideration, we feel that it has merit but does not fully meet PLOS ONE’s publication criteria as it currently stands. Therefore, we invite you to submit a revised version of the manuscript that addresses the points raised during the review process.

We look forward to receiving your revised manuscript.

Kind regards,

Ghulam Rasool

Academic Editor

PLOS ONE

Journal Requirements:

2. Please note that PLOS ONE has specific guidelines on code sharing for submissions in which author-generated code underpins the findings in the manuscript. In these cases, all author-generated code must be made available without restrictions upon publication of the work. 

Please review our guidelines at https://journals.plos.org/plosone/s/materials-and-software-sharing#loc-sharing-code and ensure that your code is shared in a way that follows best practice and facilitates reproducibility and reuse.

**Additional Editor Comments:**

Please revise the paper according to the comments of reviewers. Avoid unnecessary reference citations.

Reviewers' comments:

Reviewer's Responses to Questions

**Comments to the Author**

1. Is the manuscript technically sound, and do the data support the conclusions?

Reviewer #1: Yes

Reviewer #2: Yes

2. Has the statistical analysis been performed appropriately and rigorously? 

Reviewer #1: Yes

Reviewer #2: Yes

3. Have the authors made all data underlying the findings in their manuscript fully available?

Reviewer #1: Yes

Reviewer #2: Yes

4. Is the manuscript presented in an intelligible fashion and written in standard English?

Reviewer #1: Yes

Reviewer #2: Yes

5. Review Comments to the Author

Reviewer #1: Respect the following major corrections:

(1) Add SI units in a nomenclature table or within the text of the manuscript.

(2) The motivation part of the introduction should be improved with a clear specification of the considered novelties.

(3) Enhance the introduction with the followng recent nanofluids' studies:

Influences of Blowing and Internal Heating Processes on Steady MHD Mixed Convective Boundary Layer Flows of Radiating Titanium Dioxide-Ethylene Glycol Nanofluids

- Generation of entropy on blood conveying silver nanoparticles embedded in curved surfaces

- Efficient Passive GDQLL Scrutinization of an Advanced Steady EMHD Mixed Convective Nanofluid Flow Problem via Wakif’s-Buongiorno Approach and Generalized Transport Laws

- Further insights into mixed convective boundary layer flows of internally heated Jeffery nanofluids: Stefan's blowing case study with convective heating and thermal radiation impressions

- Influences of blowing and internal heating processes on steady MHD mixed convective boundary layer flows of radiating titanium dioxide-ethylene glycol nanofluids

- A passive modeling strategy of steady MHD reacting flows for convectively heated shear-thinning/shear-thickening nanofluids over a horizontal elongating flat surface via Wakif’s-Buongiorno approach

- Effects of fractional derivative and heat source/sink on MHD free convection flow of nanofluids in a vertical cylinder: A generalized Fourier's law model

- Exploration of Multiple Transfer Phenomena within Viscous Fluid Flows over a Curved Stretching Sheet in the Co-Existence of Gyrotactic Micro-Organisms and Tiny Particles

- Water thermal enhancement in a porous medium via a suspension of hybrid nanoparticles: MHD mixed convective Falkner's-Skan flow

- Solutal effects on thermal sensitivity of casson nanofluids with comparative investigations on Newtonian (water) and non-Newtonian (blood) base liquid

(4) The authors have the possibility of integrating the hybrid nanofluids’ concepts instead of studying a monotype nanofluid. In this respect, it is recommended to add othe the effects and replace the present flow model with a realistic one.

(5) The proposed solving strategy should be explained via a detailed flowchart and suitable references.

(6) Validate your results.

(7) Reinforce the results with proper physical explanations.

Reviewer #2: Abstract section:

The abstract is well-written and effectively communicates the objectives, methodology, and key findings of the study. However, to enhance clarity and coherence, minor revisions are suggested:

1. Specify the objective of the study more explicitly in the first sentence.

2. Clarify the novelty and significance of employing ANN analysis in triangular-shaped cavities.

3. Provide a brief explanation of the rationale behind selecting the Levenberg-Marquardt backpropagation technique for ANN.

Introduction section:

1. The introduction provides a comprehensive overview of the existing literature in the field of fluid flow and heat transfer, which is commendable. However, there are instances where the flow of information could be improved for better clarity and organization. Consider structuring the introduction into subsections to delineate different research themes or methodologies, making it easier for readers to follow the progression of ideas.

2. Ensure consistency in citation formatting throughout the introduction. Some citations are provided in brackets [1], while others are presented within the text (e.g., "Raza et al. [1]"). Adopt a consistent citation style throughout the introduction to enhance readability and professionalism.

3. Provide brief explanations of how each study contributes to the context of the current research, emphasizing their relevance to the research objectives.

4. Consider refining the articulation of this research gap to clearly convey why addressing it is important and what specific knowledge voids it aims to fill within the broader research landscape.

5. Clearly state how the identified research gap motivates the current research and what specific aspects will be addressed in the subsequent sections of the paper.

Mathematical and governing equation section:

1. It is suggested to provide description of terms such as density variation due to Boussineq approximation etc before incorporating into equation to provide readers a better understanding.

2. Also, few more steps leading to derivation of non-dimensional form could certainly benefit the readability and clarity of the paper

Conclusion section:

1. The conclusion could benefit from clearer structuring to enhance readability. Consider breaking it down into subsections corresponding to the main findings discussed, such as the impact of nanofluid concentration, forced convection, fluid velocity, and magnetohydrodynamics. This would make it easier for readers to navigate and comprehend the key findings.

2. Provide more specific details regarding the findings, such as numerical values of Nusselt numbers, temperature gradients, and fluid flow velocities for each parameter combination investigated. This would provide readers with a clearer understanding of the magnitude of the observed effects and facilitate comparison with previous studies.

3. Connect the findings discussed in the conclusion more explicitly with the existing literature reviewed in the introduction. Highlight how the observed trends and results contribute to addressing the identified research gap and advancing our understanding of fluid flow and thermal performance in triangular-shaped cavities using nanofluids and ANN analysis.

4. Discuss the broader implications of the findings in terms of their significance for practical applications and future research directions. Consider addressing how the insights gained from this study can inform the design and optimization of thermal devices and systems, as well as potential areas for further investigation.

6. PLOS authors have the option to publish the peer review history of their article (what does this mean?). If published, this will include your full peer review and any attached files.

Reviewer #1: No

Reviewer #2: No

---

## [Decision Letter · Decision Letter 1]

20 May 2024

Artificial Neural Network Analysis on the Effect of Mixed Convection in a Triangular-shaped Geometry using Water-based Al2O3 Nanofluid

PONE-D-24-08432R1

Dear Dr. Rahman,

We’re pleased to inform you that your manuscript has been judged scientifically suitable for publication and will be formally accepted for publication once it meets all outstanding technical requirements.

Kind regards,

Ghulam Rasool

Academic Editor

PLOS ONE

Additional Editor Comments (optional):

Reviewers' comments:

Reviewer's Responses to Questions

**Comments to the Author**

1. If the authors have adequately addressed your comments raised in a previous round of review and you feel that this manuscript is now acceptable for publication, you may indicate that here to bypass the “Comments to the Author” section, enter your conflict of interest statement in the “Confidential to Editor” section, and submit your "Accept" recommendation.

Reviewer #1: All comments have been addressed

Reviewer #2: All comments have been addressed

2. Is the manuscript technically sound, and do the data support the conclusions?

Reviewer #1: Yes

Reviewer #2: Yes

3. Has the statistical analysis been performed appropriately and rigorously? 

Reviewer #1: Yes

Reviewer #2: N/A

4. Have the authors made all data underlying the findings in their manuscript fully available?

Reviewer #1: Yes

Reviewer #2: Yes

5. Is the manuscript presented in an intelligible fashion and written in standard English?

Reviewer #1: Yes

Reviewer #2: Yes

6. Review Comments to the Author

Reviewer #1: The revision is well done as suggested in the first round ----------------------------------------------------------------------------------------------------------------------------------------------------------------------------------------------------------------------------------------------------------------------------------------------------------------------------------------------------------------------

Reviewer #2: (No Response)

7. PLOS authors have the option to publish the peer review history of their article (what does this mean?). If published, this will include your full peer review and any attached files.

Reviewer #1: No

Reviewer #2: No

---

## [Editor Report · Acceptance letter]

7 Jun 2024

PONE-D-24-08432R1 

PLOS ONE

Dear Dr. Rahman, 

I'm pleased to inform you that your manuscript has been deemed suitable for publication in PLOS ONE. Congratulations! Your manuscript is now being handed over to our production team.

Kind regards, 

on behalf of

Dr. Ghulam Rasool 

Academic Editor

PLOS ONE